# Offline Imitation Learning with Variational Counterfactual Reasoning

**Zexu Sun**[§]**, Bowei He**[†]**, Jinxin Liu**[‡]**, Xu Chen**[§,*]**, Chen Ma**[†]**, Shuai Zhang**[¶]

[§]Gaoling School of Artificial Intelligence, Renmin University of China
[†]Department of Computer Science, City University of Hong Kong
[‡]School of Engineering, Westlake University    [¶]DiDi Chuxing
{sunzexu21, xu.chen}@ruc.edu.cn, boweihe2-c@my.cityu.edu.hk
liujinxin@westlake.edu.cn, chenma@cityu.edu.hk, shuai.zhang@tju.edu.cn

## Abstract

In offline imitation learning (IL), an agent aims to learn an optimal expert behavior policy without additional online environment interactions. However, in many real-world scenarios, such as robotics manipulation, the offline dataset is collected from suboptimal behaviors without rewards. Due to the scarce expert data, the agents usually suffer from simply memorizing poor trajectories and are vulnerable to the variations in the environments, lacking the capability of generalizing to new environments. To automatically generate high-quality expert data and improve the generalization ability of the agent, we propose a framework named Offline Imitation Learning with Counterfactual data Augmentation (OILCA) by doing counterfactual inference. In particular, we leverage identifiable variational autoencoder to generate *counterfactual* samples for expert data augmentation. We theoretically analyze the influence of the generated expert data and the improvement of generalization. Moreover, we conduct extensive experiments to demonstrate that our approach significantly outperforms various baselines on both DEEPMIND CONTROL SUITE benchmark for in-distribution performance and CAUSALWORLD benchmark for out-of-distribution generalization.

## 1 Introduction

By utilizing the pre-collected expert data, imitation learning (IL) allows us to circumvent the difficulty in designing proper rewards for decision-making tasks and learning an expert policy. Theoretically, as long as adequate expert data are accessible, we can easily learn an imitator policy that maintains a sufficient capacity to approximate the expert behaviors [25, 30, 29, 31]. However, in practice, several challenging issues hinder its applicability to practical tasks. In particular, expert data are often limited, and due to the typical requisite for online interaction with the environment, performing such online IL may be costly or unsafe in real-world scenarios such as self-driving or industrial robotics [11, 34, 37]. Alternatively, in such settings, we might instead have access to large amounts of pre-collected unlabeled data, which are of unknown quality and may consist of both good-performing and poor-performing trajectories. For example, in self-driving tasks, a number of human driving behaviors may be available; in industrial robotics domains, one may have access to large amounts of robot data. The question then arises: can we perform offline IL with only limited expert data and the previously collected unlabeled data, thus relaxing the costly online IL requirements?

Traditional behavior cloning (BC) [5] directly mimics historical behaviors logged in offline data (both expert data and unlabeled data) via supervised learning. However, in our above setting, BC

---

*Corresponding author

37th Conference on Neural Information Processing Systems (NeurIPS 2023).

suffers from unstable training, as it relies on sufficient high-quality offline data, which is unrealistic. Besides, utilizing unlabeled data indiscriminately will lead to severe catastrophes: Bad trajectories mislead policy learning, and good trajectories fail to provide strong enough guidance signals for policy learning. In order to better distinguish the effect of different quality trajectories on policy learning, various solutions are proposed correspondingly. For example, ORIL [39] learns a reward model to relabel previously collected data by contrasting expert and unlabeled trajectories from a fixed dataset; DWBC [36] introduces a discriminator-weighted task to assist the policy learning. However, such discriminator-based methods are prone to overfitting and suffer from poor generalization ability, especially when only very limited expert data are provided.

Indeed, the expert data play an important role in offline IL and indicate the well-performing agent's intention. Thus, how to fully understand the expert's behaviors or preferences via limited available expert data becomes pretty crucial. In this paper, we propose to investigate counterfactual techniques for interpreting such expert behaviors. Specifically, we leverage the *variational counterfactual reasoning* [23, 4] to augment the expert data, as typical imitation learning data augmentation

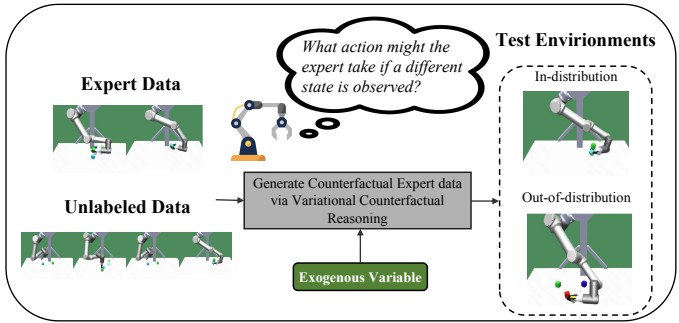

Figure 1: Agent is trained with the collected dataset containing limited expert data and large amounts of unlabeled data, and tested on both in-distribution and out-of-distribution environments.

methods [3, 7] easily generate noisy data and inevitably bias the learning agent. Here, we conduct the counterfactual reasoning of the expert by answering the following question:

*"What action might the expert take if a different state is observed?"*

Throughout this paper, we consider the structure causal model (SCM) underlying the offline training data and introduce an exogenous variable that influences the states and yet is unobserved. This variable is utilized as the minimal edits to existing expert training data so that we can generate counterfactual states that cause an agent to change its behavior. Intuitively, this exogenous variable captures variations of features in the environment. By introducing additional variations in the states during training, we encourage the model to rely less on the idiosyncrasies of a given environment. In detail, we leverage an identifiable generative model to generate the counterfactual expert data, thus enhancing the agent's capabilities of generalization in test environments (Figure 1).

The main contributions of this paper are summarized as follows:

- We propose a novel learning framework OILCA for offline IL. Using both training data and the augmentation model, we can generate counterfactual expert data and improve the generalization of the learned agent.

- We theoretically analyze the disentanglement identifiability of the constructed exogenous variable and the influence of augmented counterfactual expert data via a sampler policy. We also guarantee the improvement of generalization ability from the perspective of error bound.

- We conduct extensive experiments and provide related analysis. The empirical results about in-distribution performance on DEEPMIND CONTROL SUITE benchmark and out-of-distribution generalization on CAUSALWORLD benchmark both demonstrate the effectiveness of our method.

## 2   Related Works

**Offline IL**   A significant inspiration for this work grows from the offline imitation learning technique on how to learn policies from the demonstrations. Most of these methods take the idea of behavior cloning (BC) [5] that utilizes supervised learning to learn to act. However, due to the presence of suboptimal demonstrations, the performance of BC is limited to mediocre levels on many datasets. To address this issue, ORIL [39] learns a reward function and uses it to relabel offline trajectories.

However, it suffers from high computational costs and the difficulty of performing offline RL under distributional shifts. Trained on all data, BCND [26] reuses another policy learned by BC as the weight of the original BC objective, but its performance can even be worse if the suboptimal data occupies the major part of the offline dataset. LobsDICE [12] learns to imitate the expert policy via optimization in the space of stationary distributions. It solves a single convex minimization problem, which minimizes the divergence between the two-state transition distributions induced by the expert and the agent policy. CEIL [16] explicitly learns a hindsight embedding function together with a contextual policy. To achieve the expert matching objective for IL, CEIL advocates for optimizing a contextual variable such that it biases the contextual policy towards mimicking expert behaviors. DWBC [36] introduces an additional discriminator to distinguish expert and unlabeled demonstrations, and the outputs of the discriminator serve as the weights of the BC loss. CLUE [17] proposes to learn an intrinsic reward that is consistent with the expert intention via enforcing the embeddings of expert data to a calibrated contextual representation. OILCA aims to augment the the scarce expert data to improve the performance of the learned policy.

**Causal Dynamics RL**  Adopting this formalism allows one to cast several important problems within RL as questions of causal inference, such as off-policy evaluation [6, 22], learning baselines for model-free RL [20], and policy transfer [10, 15]. CTRL [19] applies SCM dynamics to the data augmentation in continuous sample spaces and discusses the conditions under which the generated transitions are uniquely identifiable counterfactual samples. This approach models state and action variables as unstructured vectors, emphasizing benefits in modeling action interventions for scenarios such as clinical healthcare where exploratory policies cannot be directly deployed. MOCODA [24] applies a learned locally factored dynamics model to an augmented distribution of states and actions to generate counterfactual transitions for RL. FOCUS [38] can reconstruct the causal structure accurately and illustrate the feasibility of learning causal structure in offline RL. OILCA uses the identifiable generative model to infer the distribution of the exogenous variable in the causal MDP, then performs the counterfactual data augmentation to augment the scarce expert data in offline IL.

## 3  Preliminaries

### 3.1  Problem Definition

We consider the causal Markov Decision Process (MDP) [19] with an additive noise. In our problem setting, we have an offline static dataset consisting of *i.i.d* tuples $\mathcal{D}_{\text{all}} = \left\{ s_t^i, a_t^i, s_{t+1}^i \right\}_{i=1}^{n_{\text{all}}}$ s.t. $(s_t, a_t) \sim \rho(s_t, a_t), s_{t+1} \sim f_\varepsilon(s_t, a_t, u_{t+1})$, where $\rho(s_t, a_t)$ is an offline state-action distribution resulting from some behavior policies, $f_\varepsilon(s_t, a_t, u_{t+1})$ represents the causal transition mechanism, $u_{t+1}$ is the sample of the exogenous variable $u$, which is unobserved, and $\varepsilon$ is a small permutation. Let $\mathcal{D}_E$ and $\mathcal{D}_U$ be the sets of expert and unlabeled demonstrations respectively, our goal is to only leverage the offline batch data $\mathcal{D}_{\text{all}} := \mathcal{D}_E \cup \mathcal{D}_U$ to learn an optimal policy $\pi$ without any online interaction.

### 3.2  Counterfactual Reasoning

We provide a brief background on counterfactual reasoning. Further details can be found in [23].

**Definition 1** (Structural Causal Model (SCM)). *A structural causal model $\mathcal{M}$ over variables $\mathbf{X} = \{X_1, \ldots, X_n\}$ consists of a set of independent exogenous variables $\mathbf{U} = \{\mathbf{u}_1, \ldots, \mathbf{u}_n\}$ with prior distributions $P(\mathbf{u}_i)$ and a set of functions $f_1, \ldots, f_n$ such that $X_i = f_i(\mathbf{PA}_i, \mathbf{u}_i)$, where $\mathbf{PA}_i \subset \mathbf{X}$ are parents of $X_i$. Therefore, the distribution of the SCM, which is denoted $P^{\mathcal{M}}$, is determined by the functions and the prior distributions of exogenous variables.*

Inferring the exogenous random variables based on the observations, we can intervene in the observations and inspect the consequences.

**Definition 2** (*do*-intervention in SCM). *An intervention $I = do\left(X_i := f_i\left(\tilde{\mathbf{PA}}_i, \mathbf{u}_i\right)\right)$ is defined as replacing some functions $f_i(\mathbf{PA}_i, \mathbf{u}_i)$ with $f_i\left(\tilde{\mathbf{PA}}_i, \mathbf{u}_i\right)$, where $\tilde{\mathbf{PA}}_i$ is the intervened parents of $X_i$. The intervened SCM is indicated as $\mathcal{M}^I$, and, consequently, its distribution is denoted as $P^{\mathcal{M};I}$.*

The counterfactual inference with which we can answer the *what if* questions will be obtained in the following process:

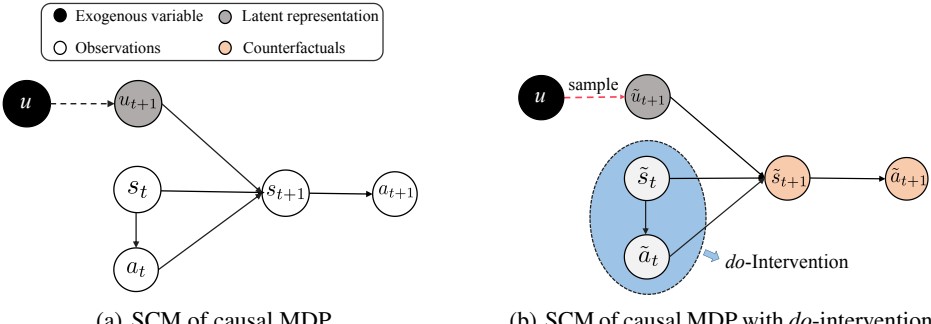

(a) SCM of causal MDP

(b) SCM of causal MDP with *do*-intervention

Figure 2: SCM of causal Markov Decision Process (MDP). We incorporate an exogenous variable in the SCM that is learned and utilized for counterfactual reasoning about *do*-intervention.

1. Infer the posterior distribution of exogenous variable $P(\mathbf{u}_i \mid \mathbf{X} = \mathbf{x})$, where $\mathbf{x}$ is a set of observations. Replace the prior distribution $P(\mathbf{u}_i)$ with the posterior distribution $P(\mathbf{u}_i \mid \mathbf{X} = \mathbf{x})$ in the SCM.

2. We denote the resulted SCM as $\mathcal{M}_{\mathbf{x}}$ and its distribution as $P^{\mathcal{M}_{\mathbf{x}}}$, perform an intervention $I$ on $\mathcal{M}_{\mathbf{x}}$ to reach $P^{\mathcal{M}_{\mathbf{x}};I}$.

3. Return the output of $P^{\mathcal{M}_{\mathbf{x}};I}$ as the counterfactual inference.

**SCM representation of causal MDP** We encode causal MDP (under a policy $\pi$) into an SCM $\mathcal{M}$. The SCM shown in Figure 2 consists of an exogenous variable $u$ and a set of functions that transmit the state $s_t$, action $a_t$ and latent representation $u_{t+1}$ of $u$ to the next state $s_{t+1}$ *e.g.* $s_{t+1} \sim \boldsymbol{f}_{\boldsymbol{\varepsilon}}(s_t, a_t, u_{t+1})$, where $\boldsymbol{\varepsilon}$ is a small perturbation, and subsequently, to the next action $a_{t+1}$, *e.g.* $a_{t+1} \sim \pi(\cdot \mid s_{t+1})$. To perform counterfactual inference on the SCM $\mathcal{M}$, we intervene in the original parents pair $(s_t, a_t)$ to $(\tilde{s}_t, \tilde{a}_t)$. Specifically, we sample the latent representation $\tilde{u}_{t+1}$ from the distribution of exogenous variable $u$, and then generate the counterfactual next state $\tilde{s}_{t+1}$, *i.e.* $\tilde{s}_{t+1} \sim \boldsymbol{f}_{\boldsymbol{\varepsilon}}(\tilde{s}_t, \tilde{a}_t, \tilde{u}_{t+1})$, and also the counterfactual next action $\tilde{a}_{t+1}$, *i.e.* $\tilde{a}_{t+1} \sim \hat{\pi}(\cdot \mid \tilde{s}_{t+1})$, where $\hat{\pi}$ is the sampler policy.

### 3.3 Variational Autoencoder and Identifiability

We briefly introduce the Variational Autoencoder (VAE) and present its identifiability results. The VAE can be conceptualized as the amalgamation of a generative latent variable model and an associated inference model, both of which are parameterized by neural networks [13]. Specifically, the VAE learns the joint distribution $p_{\boldsymbol{\theta}}(\boldsymbol{x}, \boldsymbol{z}) = p_{\boldsymbol{\theta}}(\boldsymbol{x} \mid \boldsymbol{z})p_{\boldsymbol{\theta}}(\boldsymbol{z})$, where $p_{\boldsymbol{\theta}}(\boldsymbol{x} \mid \boldsymbol{z})$ represents the conditional distribution of observing $\boldsymbol{x}$ given $\boldsymbol{z}$. Here, $\boldsymbol{\theta}$ denotes the set of generative parameters, and $p_{\boldsymbol{\theta}}(\boldsymbol{z}) = \Pi_{i=1}^{I} p_{\boldsymbol{\theta}}(z_i)$ represents the factorized prior distribution of the latent variables. By incorporating an inference model $q_{\boldsymbol{\phi}}(\boldsymbol{z} \mid \boldsymbol{x})$, the parameters $\boldsymbol{\phi}$ and $\boldsymbol{\theta}$ can be jointly optimized by maximizing the evidence lower bound (ELBO) on the marginal likelihood $p_{\boldsymbol{\theta}}(\boldsymbol{x})$.

$$
\begin{aligned}
\mathcal{L} &= \mathbb{E}_{q_{\boldsymbol{\phi}}(\boldsymbol{z}|\boldsymbol{x})}\left[\log p_{\boldsymbol{\theta}}(\boldsymbol{x} \mid \boldsymbol{z})\right] - D_{\mathrm{KL}}\left(q_{\boldsymbol{\phi}}(\boldsymbol{z} \mid \boldsymbol{x}) \| p(\boldsymbol{z})\right) \\
&= \log p_{\boldsymbol{\theta}}(\boldsymbol{x}) - D_{\mathrm{KL}}\left(q_{\boldsymbol{\phi}}(\boldsymbol{z} \mid \boldsymbol{x}) \| p_{\boldsymbol{\theta}}(\boldsymbol{z} \mid \boldsymbol{x})\right) \leq \log p_{\boldsymbol{\theta}}(\boldsymbol{x}),
\end{aligned}
\tag{1}
$$

where $D_{\mathrm{KL}}$ denotes the KL-divergence between the approximation and the true posterior, and $\mathcal{L}$ is a lower bound of the marginal likelihood $p_{\boldsymbol{\theta}}(\boldsymbol{x})$ because of the non-negativity of the KL-divergence. Recently, it has been shown that VAEs with unconditional prior distributions $p_{\boldsymbol{\theta}}(\boldsymbol{z})$ are not identifiable [18], but the latent factors $\boldsymbol{z}$ can be identified with a conditionally factorized prior distribution $p_{\boldsymbol{\theta}}(\boldsymbol{x} \mid \boldsymbol{z})$ over the latent variables to break the symmetry [9].

## 4 Offline Imitation Learning with Counterfactual Data Augmentation

At a high level, OILCA consists of following steps: (1) data augmentation via variational counterfactual reasoning, and (2) cooperatively learning a discriminator and a policy by using $\mathcal{D}_U$ and augmented $\mathcal{D}_E$. In this section, we detail these two steps and present our method's theoretical guarantees.

## 4.1 Counterfactual Data Augmentation

Our intuition is that the counterfactual expert data can model deeper relations between states and actions, which can help the learned policy be generalized better. Hence, in this section, we aim to generate counterfactual expert data by answering the *what if* question in Section 1. Consequently, counterfactual expert data augmentation is especially suitable for some real-world settings, where executing policies during learning can be too costly or slow.

As presented in Section 3.2, the counterfactual data can be generated by using the posterior of the exogenous variable obtained from the observations. Thus, we can briefly introduce the conditional VAE [13, 28] to build the latent representation of the exogenous variable $u$. Moreover, Considering the identifiability of unsupervised disentangled representation learning [18], an additionally observed variable $c$ is needed, where $c$ could be, for example, the time index or previous data points in a time series, some kind of (possibly noisy) class label, or another concurrently observed variable [9]. Formally, let $\boldsymbol{\theta} = (\boldsymbol{f}, \boldsymbol{T}, \boldsymbol{\lambda})$ be the parameters of the following conditional generative model:

$$p_{\boldsymbol{\theta}}(s_{t+1}, u \mid s_t, a_t, c) = p_{\boldsymbol{f}}(s_{t+1} \mid s_t, a_t, u) p_{\boldsymbol{T}, \boldsymbol{\lambda}}(u \mid c), \tag{2}$$

where we first define:

$$p_{\boldsymbol{f}}(s_{t+1} \mid s_t, a_t, u) = p_{\boldsymbol{\varepsilon}}(s_{t+1} - \boldsymbol{f}_{s_t, a_t}(u)). \tag{3}$$

Equation (2) describes the generative mechanism of $s_{t+1}$ given the underlying exogenous variable $u$, along with $(s_t, a_t)$. Equation (3) implies that the observed representation $s_{t+1}$ is an additive noise function, *i.e.*, $s_{t+1} = \boldsymbol{f}_{s_t, a_t}(u) + \boldsymbol{\varepsilon}$ where $\boldsymbol{\varepsilon}$ is independent of $\boldsymbol{f}_{s_t, a_t}$ or $u$. Moreover, this formulation also can be found in the SCM representation of causal MDP in Section 3.2, where we treat the function $f$ as the parameter $\boldsymbol{f}$ of the model.

Following standard conditional VAE [28] derivation, the evidence lower bound (ELBO) for each sample for $s_{t+1}$ of the above generative model can be written as:

$$\log p(s_{t+1} \mid s_t, a_t, c) \geq \mathcal{L}(\boldsymbol{\theta}, \boldsymbol{\phi}) := \log p_{\boldsymbol{f}}(s_{t+1} \mid s_t, a_t, u) + \log p(c) - \\ D_{\mathrm{KL}}(q_{\boldsymbol{\phi}}(u \mid s_t, a_t, s_{t+1}, c) \| p_{\boldsymbol{T}, \boldsymbol{\lambda}}(u \mid c)). \tag{4}$$

Note that the ELBO in Equation (4) contains an additional term of the $\log p(c)$ that does not affect identifiability but improves the estimation for the conditional prior [21], we present the theoretical guarantees of disentanglement identifiability in Section 4.3. Moreover, the encoder $q_{\boldsymbol{\phi}}$ contains all the observable variables. The reason is that, as shown in Figure 2(a), from a causal perspective, $s_t, a_t$ and $u$ form a collider at $s_{t+1}$, which means that when given $s_{t+1}$, $s_t$ and $a_t$ are related to $u$.

We seek to augment the expert data in $\mathcal{D}_E$. As shown in Figure 2(b), once the posterior of exogenous variable $q_{\boldsymbol{\phi}}(u \mid s_t, a_t, s_{t+1}, c)$ is obtained, we can perform *do*-intervention. In particular, we intervene in the parents $(s_t, a_t)$ of $s_{t+1}$ in $\mathcal{D}_E$ by re-sampling a different pair $(\tilde{s}_t, \tilde{a}_t)$ from the collected data in $\mathcal{D}_U$. Moreover, we sample the latent representation $\tilde{u}_{t+1}$ from the learned posterior distribution of exogenous variable $u$. Then utilizing $\tilde{s}_t, \tilde{a}_t$, and $\tilde{u}_{t+1}$, we can generate the counterfactual next state according to Equation (3), which is denoted as $\tilde{s}_{t+1}$. Subsequently, we pre-train a sampler policy $\hat{\pi}_E$ with the original $\mathcal{D}_E$ to sample the counterfactual next action $\tilde{a}_{t+1} = \hat{\pi}_E(\cdot \mid \tilde{s}_{t+1})$. Thus, for all the generated tuples $(\tilde{s}_{t+1}, \tilde{a}_{t+1})$, we constitute them in $\mathcal{D}_E$.

## 4.2 Offline Agent Imitation Learning

In general, the counterfactual data augmentation of our method can enhance many offline IL methods. It is worth noting that as described in [40], discriminator-based methods are easy to be over-fitted, which can more directly show the importance and effectiveness of the counterfactual augmented data. Thus, in this section, also to best leverage the unlabeled data, we use Discriminator-Weighted Behavioral Cloning (DWBC) [36], a state-of-the-art discriminator-based offline IL method. This method introduces a unified framework to learn the policy and discriminator cooperatively. The discriminator training gets information from the policy $\pi_{\boldsymbol{\omega}}$ as additional input, yielding a new discriminating task whose learning objective is as follows:

$$\mathcal{L}_{\boldsymbol{\psi}}(\mathcal{D}_E, \mathcal{D}_U) = \eta \underset{(s_t, a_t) \sim \mathcal{D}_E}{\mathbb{E}} \left[ -\log D_{\boldsymbol{\psi}}(s_t, a_t, \log \pi_{\boldsymbol{\omega}}) \right] + \underset{(s'_t, a'_t) \sim \mathcal{D}_U}{\mathbb{E}} \left[ -\log(1 - D_{\boldsymbol{\psi}}(s'_t, a'_t, \log \pi_{\boldsymbol{\omega}})) \right] \\ - \eta \underset{(s_t, a_t) \sim \mathcal{D}_E}{\mathbb{E}} \left[ -\log(1 - D_{\boldsymbol{\psi}}(s_t, a_t, \log \pi_{\boldsymbol{\omega}})) \right], \tag{5}$$

**Algorithm 1** Training procedure of OILCA.

---

**Input:** Dataset $\mathcal{D}_E$, $\mathcal{D}_U$, $\mathcal{D}_{\text{all}}$, pre-trained sampler policy $\hat{\pi}_E$, hyperparameters $\eta$, $\alpha$, initial variational couterfactual parameters $\boldsymbol{\theta}, \boldsymbol{\phi}$, discriminator parameters $\boldsymbol{\psi}$, policy parameters $\boldsymbol{\omega}$, data augmentation batch number $B$.
**Output:** Learned policy parameters $\boldsymbol{\omega}$.

  1: **while** counterfactual training **do**                       ▷ Variational counterfactual reasoning
  2:     Sample $(s_t, a_t, s_{t+1}, c) \sim \mathcal{D}_{\text{all}}$ to form a training batch
  3:     Update $\boldsymbol{\theta}$ and $\boldsymbol{\phi}$ according to Equation (4)
  4: **end while**
  5: **for** $b = 1$ to $B$ **do**                                  ▷ Expert data augmentation
  6:     Sample $(s_t, a_t, s_{t+1}, c) \sim \mathcal{D}_E$, $(\tilde{s}_t, \tilde{a}_t) \sim \mathcal{D}_U$ to form an augmentation batch
  7:     Generate the counterfactual $\tilde{s}_{t+1}$ according to Equation (2), then predict $\tilde{a}_{t+1} = \hat{\pi}_E(\cdot \mid \tilde{s}_{t+1})$
  8:     $\mathcal{D}_E \cup (\tilde{s}_{t+1}, \tilde{a}_{t+1})$
  9: **end for**
 10: **while** agent training **do**                      ▷ Learning the discriminator and policy cooperatively
 11:     Sample $(s_t, a_t) \sim \mathcal{D}_E$ and $(s'_t, a'_t) \sim \mathcal{D}_U$ to form a training batch
 12:     Update $\boldsymbol{\psi}$ according Equation (5) every 100 training steps          ▷ Discriminator learning
 13:     Update $\boldsymbol{\omega}$ according to Equation (6) every 1 training step              ▷ Policy learning
 14: **end while**

---

where $D_{\boldsymbol{\psi}}$ is the discriminator, $\eta$ is called the class prior. In the previous works [35, 39], $\eta$ is a fixed hyperparameter often assigned as 0.5.

Notice that now $\pi_{\boldsymbol{\omega}}$ appears in the input of $D_{\boldsymbol{\psi}}$, which means that imitation information from $\log \pi_{\boldsymbol{\omega}}$ will affect $\mathcal{L}_{\boldsymbol{\psi}}$, and further impact the learning of $D_{\boldsymbol{\psi}}$. Thus, inspired by the idea of adversarial training, DWBC [36] introduces a new learning objective for BC Task:

$$
\begin{aligned}
\mathcal{L}_\pi = & \alpha \mathop{\mathbb{E}}_{(s_t, a_t) \sim \mathcal{D}_E} \left[ -\log \pi_{\boldsymbol{\omega}}(a_t \mid s_t) \right] - \mathop{\mathbb{E}}_{(s_t, a_t) \sim \mathcal{D}_E} \left[ -\log \pi_{\boldsymbol{\omega}}(a_t \mid s_t) \cdot \frac{\eta}{d(1-d)} \right] \\
& + \mathop{\mathbb{E}}_{(s'_t, a'_t) \sim \mathcal{D}_U} \left[ -\log \pi_{\boldsymbol{\omega}}(a'_t \mid s'_t) \cdot \frac{1}{1-d} \right], \quad \alpha > 1.
\end{aligned}
\tag{6}
$$

where $d$ represents $D_{\boldsymbol{\psi}}(s_t, a_t, \log \pi_{\boldsymbol{\omega}})$ for simplicity, $\alpha$ is the weight factor ($\alpha > 1$). The detailed training procedure of OILCA is shown in Algorithm 1. Moreover, we also present more possible combinations with other offline IL methods and the related results in Appendix F.

### 4.3 Theoretical Analysis

In this section, we theoretically analyze our method, which mainly contains three aspects: (1) disentanglement identifiability, (2) the influence of the augmented data, and (3) the generalization ability of the learned policy.

Our disentanglement identifiability extends the theory of iVAE [9]. To begin with, some assumptions are needed. Considering the SCM in Figure 2(a), we assume the data generation process in Assumption 1.

**Assumption 1.** *(a) The distribution of the exogenous variable $u$ is independent of time but dependent on the auxiliary variable $c$. (b) The prior distributions of exogenous variable $u$ are different across auxiliary variable $c$. (c) The trasition mechanism $p(s_{t+1} \mid s_t, a_t, u)$ are invariant across different auxiliary variable $c$. (d) Given the exogenous variable $u$, the next state $s_{t+1}$ is independent of the auxiliary variable $c$. i.e. $s_{t+1} \perp\!\!\!\perp c \mid u$.*

Part of the above assumption is also used in [19]; considering the identifiability of iVAE, we also add some necessary assumptions, which are also practical. In the following discussion, we will show that when the underlying data-generating mechanism satisfies Assumption 1, the exogenous variable $u$ can be identified up to permutation and affine transformations if the conditional prior distribution $p(u|c)$ belongs to a general exponential family distribution.

**Assumption 2.** *The prior distribution of the exogenous variable $p(u|c)$ follows a general exponential family with its parameter specified by an arbitrary function $\boldsymbol{\lambda}(c)$ and sufficient statistics $\boldsymbol{T}(u) = [\boldsymbol{T_d}(u), \boldsymbol{T}_{NN}(u)]$, here $\boldsymbol{T_d}(u)$ is defined by the concatenation of $\boldsymbol{T_d}(u) =$*

$[\boldsymbol{T}_1(u^1)^T, \cdots, \boldsymbol{T}_d(u^d)^T]^T$ *from a factorized exponential family and the outputs of a neural network* $\boldsymbol{T}_{NN}(u)$ *with universal approximation power. The probability density can be written as:*

$$p_{\boldsymbol{T},\boldsymbol{\lambda}}(u \mid c) = \frac{\boldsymbol{Q}(u)}{\boldsymbol{Z}(u)} \exp\left[\boldsymbol{T}(u)^T \boldsymbol{\lambda}(c)\right]. \tag{7}$$

Under the Assumption 2, and leveraging the ELBO in Equation (4), we can obtain the following identifiability of the parameters in the model. For convenience, we omit the subscript of $\boldsymbol{f}_{s_t,a_t}$ as $\boldsymbol{f}$.

**Theorem 1.** *Assume that we observe data sampled from a generative model defined according to Equation (2)-(3) and Equation (7) with parameters $(\boldsymbol{f}, \boldsymbol{T}, \boldsymbol{\lambda})$, the following holds:*

  (i) *The set $\{s_{t+1} \in \mathcal{S} : \varphi_{\boldsymbol{\varepsilon}}(s_{t+1} = 0)\}$ has measure zero, where $\varphi_{\boldsymbol{\varepsilon}}$ is the characteristic function of the density $p_{\boldsymbol{\varepsilon}}$ defined in Equation (3).*

 (ii) *The function $\boldsymbol{f}$ is injective and all of its second-order cross partial derivatives exist.*

(iii) *The sufficient statistics $\boldsymbol{T}_{\boldsymbol{d}}$ are twice differentiable.*

(iv) *There exist $k + 1$ distinct points $c^0, \ldots, c^k$ such that the matrix*

$$L = \left(\boldsymbol{\lambda}\left(c^1\right) - \boldsymbol{\lambda}\left(c^0\right), \ldots, \boldsymbol{\lambda}\left(c^k\right) - \boldsymbol{\lambda}\left(c^0\right)\right) \tag{8}$$

*of size $k \times k$ is invertible.*

*Then, the parameters $\boldsymbol{\theta} = (\boldsymbol{f}, \boldsymbol{T}, \boldsymbol{\lambda})$ are identifiable up to an equivalence class induced by permutation and component-wise transformations.*

Theorem 1 guarantees the identifiability of Equation (2). We present its proof in Appendix A.

Moreover, Theorem 1 further implies a consistent result on the conditional VAE. If the variational distribution of encoder $q_{\boldsymbol{\phi}}$ is a broad parametric family that includes the true posterior, we have the following results.

**Theorem 2.** *Assume the following holds:*

  (i) *There exists the $(\boldsymbol{\theta}, \boldsymbol{\phi})$ such that the family of distributions $q_{\boldsymbol{\phi}}(u \mid s_t, a_t, s_{t+1}, c)$ contains $p_{\boldsymbol{\theta}}(u \mid s_t, a_t, s_{t+1}, c)$.*

 (ii) *We maximize $\mathcal{L}(\boldsymbol{\theta}, \boldsymbol{\phi})$ with respect to both $\boldsymbol{\theta}$ and $\boldsymbol{\phi}$.*

*Then, given infinite data, OILCA can learn the true parameters $\boldsymbol{\theta}^* := (\boldsymbol{f}^*, \boldsymbol{T}^*, \boldsymbol{\lambda}^*)$.*

We present the corresponding proof in Appendix B. Theorem 2 is proved by assuming our conditional VAE is flexible enough to ensure the ELBO is tight for some parameters and the optimization algorithm can achieve the global maximum of ELBO.

In our framework, the current generated expert sample pairs $(\tilde{s}_{t+1}, \tilde{a}_{t+1})$ are estimated based on the sampler policy $\hat{\pi}_E$. However, $\hat{\pi}_E$ may be not perfect, and its predicted results may contain noise. Thus, we would like to answer: "given the noise level of the sampler policy, how many samples one need to achieve sufficiently well performance?". Using $\kappa \in (0, 0.5)$ indicates the noise level of $\hat{\pi}_E$. If $\hat{\pi}_E$ can exactly recover the true action $a_{t+1}$ (i.e., $\kappa = 0$), then the generated sequences are perfect without any noise. On the contrary, $\kappa = 0.5$ means that $\hat{\pi}_E$ can only produce random results, and the generated sequences are fully noisy. Then we have the following theorem:

**Theorem 3.** *Given a hypothesis class $\mathcal{H}$, for any $\epsilon, \delta \in (0, 1)$ and $\kappa \in (0, 0.5)$, if $\hat{\pi}_E \in \mathcal{H}$ is the pretrained policy model learned based on the empirical risk minimization (ERM), and the sample complexity (i.e., number of samples) is larger than $\frac{2 \log\left(\frac{2\mathcal{H}}{\delta}\right)}{\epsilon^2 (1-2\kappa)^2}$, then the error between the model estimated and true results is smaller than $\epsilon$ with probability larger than $1 - \delta$.*

The related proof details are presented in Appendix C. From Theorem 3, we can see: in order to guarantee the same performance with a given probability (i.e., $\epsilon$ and $\delta$ are fixed), one needs to generate more than $\frac{2 \log\left(\frac{2\mathcal{H}}{\delta}\right)}{\epsilon^2 (1-2\kappa)^2}$ sequences. If the noise level $\kappa$ is larger, more samples have to be generated. Extremely, if the pre-trained policy can only produce fully noisy information (i.e., $\kappa = 0.5$), then infinity number of samples are required, which is impossible in reality.

For the generalization ability, [8] explains the efficacy of counterfactual augmented data by the empirical evidence. In the context of offline IL, a straightforward yet very relevant conclusion from the analysis of generalization ability is the strong dependence on the number of expert data [25]. We work with finite state and action spaces ($|\mathcal{S}|, |\mathcal{A}| < \infty$), and for the learned policy $\pi_{\omega}$, we can analyze the generalization ability from the perspective of error upper bound with the optimal expert policy $\pi^*$.

**Theorem 4.** *Let $|\mathcal{D}_E|$ be the number of empirical expert data used to train the policy and $\epsilon$ be the expected upper bound of generalization error. There exists constant $h$ such that, if*

$$|\mathcal{D}_E| \geq \frac{h|\mathcal{S}||\mathcal{A}|\log(|\mathcal{S}|/\delta)}{\epsilon^2}, \tag{9}$$

*and each state $s_t$ is sampled uniformly, then, with probability at least $1 - \delta$, we have:*

$$\max_{s_t} \|\pi^*(\cdot \mid s_t) - \pi_{\omega}(\cdot \mid s_t)\|_1 \leq \epsilon. \tag{10}$$

*which shows that increasing $|\mathcal{D}_E|$ drastically improves the generalization guarantee.*

Note that we provide the proof details for Theorem 4 in Appendix D.

## 5 Experiments

In this section, we evaluate the performance of OILCA, aiming to answer the following questions. **Q1:** With the synthetic toy environment of the causal MDP, can OILCA disentangle the latent representations of the exogenous variable? **Q2:** For an in-distribution test environment, can OILCA improve the performance? **Q3:** For an out-of-distribution test environment, can OILCA improve the generalization? In addition, we use five baseline methods: BC-exp (BC on the expert data $\mathcal{D}_E$), BC-all (BC on all demonstrations $\mathcal{D}_{\text{all}}$), ORIL [39], BCND [26], LobsDICE [12], DWBC [36]. More details about these methods are presented in Appendix E. Furthermore, the hyper-parameters of our method and baselines are all detail-tuned for better performance.

### 5.1 Simulations on Toy Environment (Q1)

In real-world situations, we can hardly get the actual distributions of the exogenous variable. To evaluate the disentanglement identifiability in variational counterfactual reasoning, we build a toy environment to show that our method can indeed get the disentangled distributions of the exogenous variable.

**Toy environment** For the toy environment, we consider a simple 2D navigation task. The agent can move a fixed distance in each of the four directions. States are continuous and are considered as the 2D position of the agent. The goal is to navigate to a specific target state within a bounding box. The reward is the negative distance between the agent's state and the target state. For the initialization of the environment, we consider $C$ kinds of Gaussian distributions (Equation (7)) for the exogenous variable, where $C = 3$, and the conditional variable $c$ is the class label. We design the transition function by using a multi-layer perceptron (MLP) with invertible activations. We present the details about data collection and model learning in Appendix E.

**Results** We visualize the identifiability in such a 2D case in Figure 3, where we plot the sources, the posterior distributions learned by OILCA, and the posterior distributions learned by a vanilla conditional VAE, respectively. Our method recovers the original sources to trivial indeterminacies (rotation and sign flip), whereas the vanilla conditional VAE fails to separate the latent variables well. To show the effectiveness of our method in the constructed toy environment, we also conduct repeated experiments for 1K episodes (500 steps per episode) to compare OILCA with all baseline methods. The results are presented in Figure 4, showing that OILCA achieves the highest average return. To analyze the influence of the number of augmented samples (Theorem 4), we also conduct the experiments with varying numbers of counterfactual expert data; the result is shown in Figure 5. The X-axis represents the percentage of $|\mathcal{D}_E|/|\mathcal{D}_U|$, where $\mathcal{D}_E$ is the augmented expert data. We can observe that within a certain interval, the generalization ability of the learned policy does improve with the increase of $|\mathcal{D}_E|$.

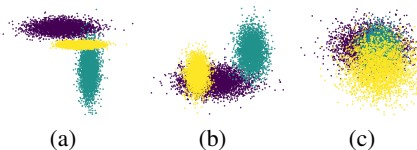
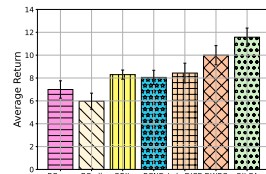
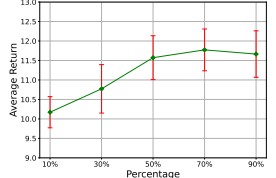

(a)       (b)       (c)

Figure 3: Visualization of both observation and latent spaces of the exogenous variable. (a) Samples from the true distribution of the sources $p_{\theta^*}(u|c)$. (b) Samples from the posterior $q_{\phi}(u|s_t, a_t, s_{t+1}, c)$. (c) Samples from the posterior $q_{\phi'}(u|s_t, a_t, s_{t+1})$ without class label.

Figure 4: Performance of OILCA and baselines in the toy environment. We plot the mean and the standard errors of the averaged return over five random seeds.

Figure 5: Performance of OILCA with the growing percentage of $|\mathcal{D}_E|/|\mathcal{D}_U|$. We plot the average return's mean and standard errors over five random seeds.

Table 1: Results for in-distribution performance on DEEPMIND CONTROL SUITE. We report the average return of episodes (with the length of 1K steps) over five random seeds. The best results and second best results are **bold** and underlined, respectively.

| Task Name | BC-exp | BC-all | ORIL | BCND | LobsDICE | DWBC | **OILCA** |
|---|---|---|---|---|---|---|---|
| CARTPOLE SWINGUP | 195.44 ± 7.39 | 269.03 ± 7.06 | 221.24 ± 14.49 | 243.52 ± 11.33 | 292.96 ± 11.05 | 382.55 ± 8.95 | **608.38** ± 35.54 |
| CHEETAH RUN | 66.59 ± 11.09 | 90.01 ± 31.74 | 45.08 ± 9.88 | 96.06 ± 16.15 | 74.53 ± 7.75 | 66.87 ± 4.60 | **116.05** ± 14.65 |
| FINGER TURN HARD | 129.20 ± 4.51 | 104.56 ± 8.32 | 185.57 ± 26.75 | 204.67 ± 13.18 | 190.93 ± 12.19 | 243.47 ± 17.12 | **298.73** ± 5.11 |
| FISH SWIM | 74.59 ±11.73 | 68.87 ± 11.93 | 84.90 ±1.96 | 153.28 ±19.29 | 188.84 ±11.28 | 212.39 ±7.62 | **290.28** ± 10.07 |
| HUMANOID RUN | 77.59 ±8.63 | 138.93 ±9.14 | 96.88 ±10.76 | 257.01 ±11.21 | 120.87 ±10.66 | 302.33 ±14.53 | **460.96** ±17.55 |
| MANIPULATOR INSERT BALL | 91.61 ±7.49 | 98.59 ±1.62 | 98.25 ±2.68 | 141.71 ±15.30 | 197.79 ±1.98 | 107.86 ±15.01 | **296.83** ±3.43 |
| MANIPULATOR INSERT PEG | 92.02 ±15.04 | 119.63 ±6.37 | 105.86 ±5.10 | 220.66 ±15.14 | 299.19 ±3.40 | 238.39 ±19.76 | **305.66** ±4.91 |
| WALKER STAND | 169.14 ± 8.00 | 192.14 ± 37.91 | 181.23 ± 10.31 | 279.66 ± 12.69 | 252.34 ± 7.73 | 280.07 ± 5.79 | **395.51** ± 8.05 |
| WALKER WALK | 29.44 ± 2.18 | 75.79 ± 6.34 | 41.43 ± 4.05 | 157.44 ± 9.31 | 102.14 ± 5.94 | 166.95 ± 10.68 | **377.19** ± 15.87 |

## 5.2 In-distribution Performance (Q2)

We use DEEPMIND CONTROL SUITE [32] to evaluate the performance of in-distribution performance. Similar to [39], we also define an episode as positive if its episodic reward is among the top 20% episodes for the task. For the auxiliary variable $c$, we add three kinds of different Gaussian noise distributions into the environment (encoded as $c = \{0, 1, 2\}$). Moreover, we present the detailed data collection and statistics in Appendix E.1). We report the results in Table 1. We can observe that OILCA outperforms baselines on all tasks; this shows the effectiveness of the augmented counterfactual expert data. We also find that the performance of ORIL and DWBC tends to decrease in testing for some tasks (ORIL: CHEETAH RUN, WALKER WALK; DWBC: CHEETAH RUN, MANIPULATOR INSERT BALL); this "overfitting" phenomenon also occurs in experiments of previous offline RL works [14, 33]. This is perhaps due to limited data size and model generalization bottleneck. DWBC gets better results on most tasks, but for some tasks, such as Manipulator Insert Ball and the Walker Walk, OILCA achieves more than twice the average return than DWBC.

## 5.3 Out-of-distribution Generalization (Q3)

To evaluate the out-of-distribution generalization of OILCA, we use a benchmark named CAUSAL-WORLD [2], which provides a combinatorial family of tasks with a common causal structure and

Table 2: Results for out-of-distribution generalization on CAUSALWORLD. We report the average return of episodes (length varies for different tasks) over five random seeds. All the models are trained on *space* **A** and tested on *space* **B** to show the out-of-distribution performance [2].

| Task Name | BC-exp | BC-all | ORIL | BCND | LobsDICE | DWBC | **OILCA** |
|---|---|---|---|---|---|---|---|
| REACHING | 281.18 | 176.54 | 339.40 | 228.33 | 243.29 | 479.92 | **976.60** |
| | ± 16.45 | ± 9.75 | ± 12.98 | ± 7.14 | ± 9.84 | ± 18.75 | ± 20.13 |
| PUSHING | 256.64 | 235.58 | 283.91 | 191.23 | 206.44 | 298.09 | **405.08** |
| | ± 12.70 | ± 10.23 | ± 19.72 | ± 12.64 | ± 15.35 | ± 14.94 | ± 24.03 |
| PICKING | 270.01 | 258.54 | 388.15 | 221.89 | 337.78 | 366.26 | **491.09** |
| | ± 13.13 | ± 16.53 | ± 19.21 | ± 7.68 | ± 12.09 | ± 8.77 | ± 6.44 |
| PICK AND PLACE | 294.06 | 225.42 | 270.75 | 259.12 | 266.09 | 349.66 | **490.24** |
| | ± 7.34 | ± 12.44 | ± 14.87 | ± 8.01 | ± 10.31 | ± 7.39 | ± 11.69 |
| STACKING2 | 496.63 | 394.91 | 388.55 | 339.18 | 362.47 | 481.07 | **831.82** |
| | ± 7.68 | ± 16.98 | ± 10.93 | ± 9.46 | ± 17.05 | ± 10.11 | ± 11.78 |
| TOWERS | 667.81 | 784.88 | 655.96 | 139.30 | 535.74 | 768.68 | **994.82** |
| | ± 9.27 | ± 17.17 | ± 15.14 | ± 18.22 | ± 13.76 | ± 24.77 | ± 5.76 |
| STACKED BLOCKS | 581.91 | 452.88 | 702.15 | 341.97 | 250.44 | 1596.96 | **2617.71** |
| | ± 26.92 | ± 18.78 | ± 15.30 | ± 33.69 | ± 14.08 | ± 81.84 | ± 88.07 |
| CREATIVE STACKED BLOCKS | 496.32 | 529.82 | 882.27 | 288.55 | 317.95 | 700.23 | **1348.49** |
| | ± 26.92 | ± 31.01 | ± 46.79 | ± 19.63 | ± 32.03 | ± 13.71 | ± 55.05 |
| GENERAL | 492.78 | 547.32 | 647.95 | 195.06 | 458.27 | 585.98 | **891.14** |
| | ±17.64 | ±8.49 | ±24.39 | ±9.80 | ±18.69 | ±19.25 | ±23.12 |

underlying factors. Furthermore, the auxiliary variable $c$ represents the different *do*-interventions on the task environments [2]. We collect the offline data by using three different *do*-interventions on environment features (`stage_color`, `stage_friction`. `floor_friction`) to generate offline datasets, while other features are set as the defaults. More data collection and statistics details are presented in Appendix E.1. We show the comparative results in Table 2. It is evident that OILCA also achieves the best performance on all the tasks; ORIL and DWBC perform the second-best results on most tasks. BCND performs poorly compared to other methods. The reason may be that the collected data in space **A** can be regarded as low-quality data when evaluating on space **B**, which may lead to the poor performance of a single BC and cause cumulative errors for the BC ensembles of BCND. LobsDICE even performs poorer than BC-exp on most tasks. This is because the KL regularization, which regularizes the learned policy to stay close to $\mathcal{D}_U$ is too conservative, resulting in a suboptimal policy, especially when $\mathcal{D}_U$ contains a large collection of noisy data. This indeed hurts the performance of LobsDICE for out-of-distribution generalization.

## 6 Conclusion

In this paper, we propose an effective and generalizable offline imitation learning framework OILCA, which can learn a policy from the expert and unlabeled demonstrations. We apply a novel identifiable counterfactual expert data augmentation approach to facilitate the offline imitation learning. We also analyze the influence of generated data and the generalization ability theoretically. The experimental results demonstrate that our method achieves better performance in simulated and public tasks.

## Acknowledgement

This work is supported in part by National Key R&D Program of China (2022ZD0120103), National Natural Science Foundation of China (No. 62102420), Beijing Outstanding Young Scientist Program NO. BJJWZYJH012019100020098, Intelligent Social Governance Platform, Major Innovation & Planning Interdisciplinary Platform for the "Double-First Class" Initiative, Renmin University of China, Public Computing Cloud, Renmin University of China, fund for building world-class universities (disciplines) of Renmin University of China, Intelligent Social Governance Platform.

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
