# A  Proof of Theorem 1

In this section, we provide proof for the disentanglement identifiability of the inferred exogenous variable. Our proof consists of three main components. It is worth noting that we also use $\boldsymbol{f}$ to replace $\boldsymbol{f}_{s_t,a_t}$ for simplicity.

*Proof.* The following are the three steps:

**Step I** We use the first assumption in Theorem 1 to demonstrate that the observed data distributions are equivalent to the noiseless distributions. Specifically, suppose that we have two sets of parameters $(\boldsymbol{f}, \boldsymbol{T}, \boldsymbol{\lambda})$ and $(\tilde{\boldsymbol{f}}, \tilde{\boldsymbol{T}}, \tilde{\boldsymbol{\lambda}})$, such that for all pairs $(s_{t+1}, c)$ $((s, c)$ for simplicity), we have:

$$\tilde{p}_{\boldsymbol{T},\boldsymbol{\lambda},\boldsymbol{f},c}(s) = \tilde{p}_{\tilde{\boldsymbol{T}},\tilde{\boldsymbol{f}},\tilde{\boldsymbol{\lambda}},c}(s) \tag{11}$$

$$p_{\boldsymbol{\theta}}(s \mid c) = p_{\tilde{\boldsymbol{\theta}}}(s \mid c) \tag{12}$$

$$\implies \int p_{\boldsymbol{\varepsilon}}(s - \boldsymbol{f}(u)) p_{\boldsymbol{T},\boldsymbol{\lambda}}(u \mid c) du = \int p_{\boldsymbol{\varepsilon}}(s - \tilde{\boldsymbol{f}}(u)) p_{\tilde{\boldsymbol{T}},\tilde{\boldsymbol{\lambda}}}(u \mid c) du \tag{13}$$

$$\implies \int p_{\boldsymbol{\varepsilon}}(s - \overline{s}) p_{\boldsymbol{T},\boldsymbol{\lambda}}\left(\boldsymbol{f}^{-1}(\overline{s} \mid c)\right) \mathrm{vol}\left(J_{f^{-1}}(\overline{s})\right) d\overline{s} = \int p_{\boldsymbol{\varepsilon}}(s - \overline{s}) p_{\boldsymbol{T},\boldsymbol{\lambda}}\left(\tilde{\boldsymbol{f}}^{-1}(\overline{s} \mid c)\right) \mathrm{vol}\left(J_{\tilde{f}^{-1}}(\overline{s})\right) d\overline{s} \tag{14}$$

$$\implies \int p_{\boldsymbol{\varepsilon}}(s - \overline{s}) \tilde{p}_{\boldsymbol{T},\boldsymbol{\lambda},\boldsymbol{f},c}(\overline{s}) d\overline{s} = \int p_{\boldsymbol{\varepsilon}}(s - \overline{s}) \tilde{p}_{\tilde{\boldsymbol{T}},\tilde{\boldsymbol{f}},\tilde{\boldsymbol{\lambda}},c}(\overline{s}) d\overline{s} \tag{15}$$

$$\implies \left(\tilde{p}_{\boldsymbol{T},\boldsymbol{\lambda},\boldsymbol{f},c} * p_{\boldsymbol{\varepsilon}}\right)(s) = \left(\tilde{p}_{\tilde{\boldsymbol{T}},\tilde{\boldsymbol{f}},\tilde{\boldsymbol{\lambda}},c} * p_{\boldsymbol{\varepsilon}}\right)(s) \tag{16}$$

$$\implies F\left[\tilde{p}_{\boldsymbol{T},\boldsymbol{\lambda},\boldsymbol{f},c}\right](\boldsymbol{\omega}) \varphi_{\boldsymbol{\varepsilon}}(\boldsymbol{\omega}) = F\left[\tilde{p}_{\tilde{\boldsymbol{T}},\tilde{\boldsymbol{f}},\tilde{\boldsymbol{\lambda}}}\right](\boldsymbol{\omega}) \varphi_{\boldsymbol{\varepsilon}}(\boldsymbol{\omega}) \tag{17}$$

$$\implies F\left[\tilde{p}_{\boldsymbol{T},\boldsymbol{\lambda},\boldsymbol{f},c}\right](\boldsymbol{\omega}) = F\left[\tilde{p}_{\tilde{\boldsymbol{T}},\tilde{\boldsymbol{f}},\tilde{\boldsymbol{\lambda}},c}\right](\boldsymbol{\omega}) \tag{18}$$

$$\implies \tilde{p}_{\boldsymbol{T},\boldsymbol{\lambda},\boldsymbol{f},c}(s) = \tilde{p}_{\tilde{\boldsymbol{T}},\tilde{\boldsymbol{f}},\tilde{\boldsymbol{\lambda}},c}(s). \tag{19}$$

where:

- in Equation (14), $J$ denotes the Jacobian, and we make the change of variable $\overline{s} = \boldsymbol{f}(u)$ on the left-hand side, and $\overline{s} = \tilde{\boldsymbol{f}}(u)$ on the right-hand side.

- in Equation (15), we introduce

$$\tilde{p}_{\boldsymbol{T},\boldsymbol{\lambda},\boldsymbol{f},c} \triangleq p_{\boldsymbol{T},\boldsymbol{\lambda}}\left(\boldsymbol{f}^{-1}\right)(s \mid c) \mathrm{vol}\left(J_{f^{-1}}(s)\right) \mathbb{I}(s) \tag{20}$$

- in Equation (16), $*$ denotes the convolution operator.

- in Equation (17), $F$ denotes the Fourier transformation and $\varphi_{\boldsymbol{\varepsilon}} = F\left[p_{\boldsymbol{\varepsilon}}\right]$.

- in Equation (18), $\varphi_{\boldsymbol{\varepsilon}}(\boldsymbol{w})$ is dropped because it is non-zero almost everywhere according to the first assumption of Theorem 1.

Equation (19) is valid for all $(s, c)$. What is basically says is that for the distributions to be the same after adding the noise, the noise-free distributions have to be the same. Note that $s$ here is a general variable, and we are actually dealing with the noise-free probability densities.

**Step II** Using Equation (20) to substitute Equation (19), we have

$$p_{\boldsymbol{T},\boldsymbol{\lambda}}\left(\boldsymbol{f}^{-1}\right)(s \mid c) \mathrm{vol}\left(J_{f^{-1}}(s)\right) \mathbb{I}(s) = p_{\tilde{\boldsymbol{T}}},\tilde{\boldsymbol{\lambda}}\left(\tilde{\boldsymbol{f}}^{-1}\right)(s \mid c) \mathrm{vol}\left(J_{\tilde{f}^{-1}}(s)\right) \mathbb{I}(s). \tag{21}$$

Then, we can apply logarithm on the above equation and substitute $p_{\boldsymbol{T},\boldsymbol{\lambda}}$ with its definition in Equation (3), and obtain

$$\log \mathrm{vol}\left(J_{f^{-1}}(s)\right) \log Q\left(\boldsymbol{f}^{-1}s\right) - \log Z(c) + \left\langle \boldsymbol{T}\left(\boldsymbol{f}^{-1}(s)\right), \boldsymbol{\lambda}(c)\right\rangle$$

$$= \log \mathrm{vol}\left(J_{\tilde{f}^{-1}}(s)\right) \log \tilde{Q}\left(\tilde{\boldsymbol{f}}^{-1}s\right) - \log \tilde{Z}(c) + \left\langle \tilde{\boldsymbol{T}}\left(\tilde{\boldsymbol{f}}^{-1}(s)\right), \tilde{\boldsymbol{\lambda}}(c)\right\rangle \tag{22}$$

Let $c^0, \cdots, c^k$ be the $k+1$ points defined in the fourth assumption of Theorem 1, we can obtain $k+1$ equation. By subtracting the first equation from the remaining $k$ equations, we then obtain:

$$\left\langle T\left(f^{-1}(s)\right), \lambda\left(c^l\right) - \lambda\left(c^0\right)\right\rangle + \log\frac{Z\left(c^0\right)}{Z\left(c^l\right)}$$

$$= \left\langle \tilde{T}\left(\tilde{f}^{-1}(s)\right), \tilde{\lambda}\left(c^l\right) - \tilde{\lambda}\left(c^0\right)\right\rangle + \log\frac{\tilde{Z}\left(c^0\right)}{\tilde{Z}\left(c^l\right)}, \qquad (23)$$

where $l = 1, \cdots, k$. Let $b \in \mathbb{R}^k$ in which $b_l = \log\frac{\tilde{Z}(c^0)Z(c^l)}{\tilde{Z}(c^l)Z(c^0)}$, we have

$$L^T T\left(f^{-1}(s)\right) = \tilde{L}\tilde{T}\left(\tilde{f}^{-1}(s)\right) + m \qquad (24)$$

Finally, we multiply both side by $L^{-T}$ and obtain

$$T\left(f^{-1}(s)\right) = A\tilde{T}\left(\tilde{f}^{-1}(s)\right) + n. \qquad (25)$$

where $A = L^{-T}L$ and $n = L^{-T}m$.

**Step III** Now recall the definition of $T$ and the third assumption. We start by evaluating Equation (25) at $k+1$ points of $u^l, s^l$ and obtain $k+1$ equations. Then, we subtract the first equation from the remaining $k+1$ equations:

$$\left[T\left(u_1\right) - T\left(u^0\right), \cdots, T\left(u^k\right) - T\left(u^0\right)\right]$$
$$= A\left[\tilde{T}\left(\tilde{f}^{-1}\left(s^1\right)\right) - \tilde{T}\left(\tilde{f}^{-1}\left(s^0\right)\right), \cdots, \tilde{T}\left(\tilde{f}^{-1}\left(s^l\right)\right) - \tilde{T}\left(\tilde{f}^{-1}\left(s^0\right)\right)\right]. \qquad (26)$$

Next, we only need to show that for $u_0$, there exist $k$ points $u^1, \cdots, u^k$ such that the columns are linear independent, which can be proven by contradiction. Suppose that there exists no such $u^l \in \{u^0, \cdots, u^k\}$, then $\left\langle T(u^l) - T\left(u^0\right), \lambda\right\rangle = 0$ and thus $T(u^l) = T\left(u^0\right) = \text{const}$. This contradicts with the assumption that the prior distribution is strongly exponential. Therefore, there must exist $k+1$ points such that the transformation is invertible. Then we have $(f, T, \lambda) \sim (\tilde{f}, \tilde{T}, \tilde{\lambda})$. $\qquad \square$

# B  Proof of Theorem 2

According to Equation (4), if the family $q_\phi\left(u \mid s_t, a_t, s_{t+1}, c\right)$ is large enough to include $p_\theta\left(u \mid s_{t+1}, s_t, a_t, c\right)$, then by optimizing the loss over its parameter $\phi$, we will minimize the KL term, eventually reaching zero, and the loss will be equal to the log-likelihood.

The conditional VAE, in this case, inherits all the properties of maximum likelihood estimation. In this particular case, since our identifiability is guaranteed up to equivalence classes, the consistency of MLE means that we converge to the equivalence class (Theorem 1) of true parameter $\theta^*$ *i.e.* Under the condition of infinite data.

# C  Proof of Theorem 3

Suppose the prediction error of $\hat{\pi}_E$ is $e$ (*i.e.*, $\sum \mathbb{I}(a_t^{\hat{\pi}_E} \neq a_t) = e$), $a_t$ is the true action that an expert take, then the mismatching probability between the observed and predicted results comes from two parts: (1) The observed result is true, but the prediction is wrong, that is, $e(1 - \kappa)$. (2) The observed result is wrong, but the prediction is right, that is $(1 - e)\kappa$. Thus, the total mismatching probability is $\kappa + e(1 - 2\kappa)$.

The following proof is based on the reduction to absurdity. We first propose an assumption and then derive contradicts to invalidate the assumption.

**Assumption.** Suppose the prediction error of $\hat{\pi}_E$ (i.e., $e$) is larger than $\epsilon$. Then, at least one of the following statements hold:

(1)  The empirical mis-matching rate of $\hat{\pi}_E$ is smaller than $\kappa + \frac{\epsilon(1-2\kappa)}{2}$.

(2) The empirical mis-matching rate of the optimal $h^* \in \mathcal{H}$ (*i.e.*, the prediction error of $h^*$ is 0) is larger than $\kappa + \frac{\epsilon(1-2\kappa)}{2}$.

These statements are easy to understand, since if both of them do not hold, we can conclude that the empirical loss of $\hat{\pi}_E$ is larger than that of $h^*$, which does not agree with the ERM definition.

**Contradicts.** To begin with, we review the uniform convergence properties [27] by the following lemma:

**Lemma 1.** *Let $\mathcal{H}$ be a hypothesis class, then for any $\epsilon \in (0,1)$ and $h \in \mathcal{H}$, if the number of training samples is $m$, the following formula holds:*

$$\mathbb{P}(|R(h) - \hat{R}(h)| > \epsilon) < 2|\mathcal{H}| \exp\left(-2m\epsilon^2\right)$$

*where $R$ and $\hat{R}$ are the expectation and empirical losses, respectively.*

For statement (1), since the prediction error of $\hat{\pi}_E$ is larger than $\epsilon$, the expectation loss $R(\hat{\pi}_E)$ is larger than $\kappa + \epsilon(1-2\kappa)$. If the empirical loss $\hat{R}(\hat{\pi}_E)$ is smaller than $\kappa + \frac{\epsilon(1-2\kappa)}{2}$, then $|R(\hat{\pi}_E) - \hat{R}(\hat{\pi}_E)|$ should be larger than $\frac{\epsilon(1-2\kappa)}{2}$. At the same time, according to Lemma 1, when the sample number $m$ is larger than $\frac{2\log\left(\frac{2|\mathcal{H}|}{\delta}\right)}{\epsilon^2(1-2\kappa)^2}$, we have $\mathbb{P}\left(|R(\hat{\pi}_E) - \hat{R}(\hat{\pi}_E)| > \frac{\epsilon(1-2\kappa)}{2}\right) < \delta$.

For statement (2), the expectation loss of $h^*$ is $\kappa$, i.e., $R(h^*) = \kappa$. If the empirical loss $\hat{R}(h^*)$ is larger than $\kappa + \frac{\epsilon(1-2\kappa)}{2}$, then $|R(h^*) - \hat{R}(h^*)|$ should be larger than $\frac{\epsilon(1-2\kappa)}{2}$. According to Lemma 1, when the sample number $m$ is larger than $\frac{2\log\left(\frac{2|\mathcal{H}|}{\delta}\right)}{\epsilon^2(1-2\kappa)^2}$, we have $\mathbb{P}\left(|R(h^*) - \hat{R}(h^*)| > \frac{\epsilon(1-2\kappa)}{2}\right) < \delta$.

As a result, both of the above statements hold with the probability smaller than $\delta$, which implies that the prediction error of $\hat{\pi}_E$ is smaller than $\epsilon$ with the probability larger than $1 - \delta$.

# D  Proof of Theorem 4

**Lemma 2.** *(Proposition A.8 of Agarwal et al. [1]). Let $z$ be a discrete random variable that takes values in $\{1, \ldots, d\}$, distributed according to $q$. We write $q$ as a vector where $\vec{q} = [\Pr(z = j)]_{j=1}^d$. Assume we have $n$ i.i.d. samples, and that our empirical estimate of $\vec{q}$ is $[\hat{q}]_j = \sum_{i=1}^n \mathbf{1}[z_i = j]/n$. We have that $\forall \epsilon > 0$ :*

$$\Pr\left(\|\hat{q} - \vec{q}\|_2 \ge 1/\sqrt{n} + \epsilon\right) \le e^{-n\epsilon^2}$$

*which implies that:*

$$\Pr\left(\|\hat{q} - \vec{q}\|_1 \ge \sqrt{d}(1/\sqrt{n} + \epsilon)\right) \le e^{-n\epsilon^2}$$

*Proof.* Applying Lemma 2, we have that for considering a fixed $s_t$, wp. at least $1 - \delta$:

$$\|\pi(\cdot \mid s_t) - \pi_{\boldsymbol{\omega}}(\cdot \mid s_t)\|_1 \le h\sqrt{\frac{|\mathcal{A}|\log(1/\delta)}{n}} \tag{27}$$

where $n$ is the number of expert data used to estimate $\pi_{\boldsymbol{\omega}}(\cdot \mid s_t)$. Then we apply the union bound across all states and actions to get that wp. at least $1 - \delta$:

$$\max_{s_t} \|\pi(\cdot \mid s_t) - \pi_{\boldsymbol{\omega}}(\cdot \mid s_t)\|_1 \le h\sqrt{\frac{|\mathcal{S}||\mathcal{A}|\log(|\mathcal{S}|/\delta)}{n}} \tag{28}$$

The result follows by rearranging $n$ and relabeling $h$. $\square$

**Remark 1.** *How much counterfactual expert data can we generate using our OILCA framework? Supposing we have $n$ independent state action tuples in the expert data, we run the data augmentation module for $m$ times, which means that we can augment each state to $m$ counterfactual states and subsequently to $m$ corresponding counterfactual actions. Thus, in total, we can obtain $n^m$ counterfactual tuples–an exponential increase for the previously given expert data. Back to Theorem 4, this demonstrates that our OILCA can effectively enhance the policy's generalization ability.*

# E Training Details

## E.1 Data Generation and Statistics

**Toy Environment** The dimensions of state and action are both 2. For the exogenous variable, we generate the non-stationary 2D Gaussian data as follows: $u^* \mid c \sim \mathcal{N}\left(\mu(c), \mathrm{diag}\left(\sigma^2(c)\right)\right)$, where $c$ is the class label. $\mu_1(c) = 0$ for all $c$ and $\mu_2(c) = \alpha\gamma(c)$, where $\alpha \in \mathbb{R}$ and $\gamma$ is a permutation. The variance $\sigma^2(c)$ is generated randomly and independently across the classes. For the transition function, we use an MLP to generate the next state $s_{t+1}$, such that $s_{t+1} = \mathrm{MLP}(s_t, a_t, u_{t+1})$, where $u_{t+1}$ is the sample of $u$ at timestep $t + 1$. For each class of exogenous variables, we generate 1K episodes for the data collection (500 steps per episode). Similar to DEEPMIND CONTROL SUITE, we also define a positive episode if its reward is among the top 20% episodes, and each of these positives is randomly chosen to constitute $\mathcal{D}_E$ with $\frac{1}{10}$ chance. As a result, we choose 75 episodes in $\mathcal{D}_E$ and 925 episodes in $\mathcal{D}_U$. For the online testing, we can evaluate all the methods on the toy environment with any kind of distribution of the exogenous variable.

**DEEPMIND CONTROL SUITE** DEEPMIND CONTROL SUITE (Figure 6) contains a variety of continuous control tasks involving locomotion and simple manipulation. States consist of joint angles and velocities, and action spaces vary depending on the task. The episodes are 1000 steps long, and the environment reward is continuous, with a maximum value of 1 per step. During the collection of offline data, we apply random Gaussian perturbation to the action outputted by the policy. This perturbation is specified in the XML configuration file as an integral part of the environment. Additionally, the distribution of the perturbation differs across different environment initialization (auxiliary variable $c$) due to their initialization seeds. In particular, different seeds correspond to different mean and variance of the Gaussian distribution perturbation via the random number generator. This approach is employed to introduce uncertainty into the environment [], thereby aligning with our problem setting. We define an episode as positive if its episodic return is among the top 20% episodes; each of these positives is randomly chosen to constitute $\mathcal{D}_E$ with $\frac{1}{10}$ chance. We present the details in Table 3.

| Task | Total | $\mathcal{D}_E$ |
|------|-------|------|
| Cartpole swingup | 40 | 2 |
| Cheetah run | 300 | 3 |
| Finger turn hard | 500 | 9 |
| Fish swim | 200 | 1 |
| Humanoid run | 3000 | 53 |
| Manipulator insert ball | 1500 | 30 |
| Manipulator insert peg | 1500 | 23 |
| Walker stand | 200 | 4 |
| Walker walk | 200 | 6 |
| Reaching | 600 | 12 |
| Pushing | 600 | 13 |
| Picking | 600 | 15 |
| Pick and Place | 600 | 12 |
| Stacking2 | 600 | 11 |
| Towers | 600 | 13 |
| Stacked Blocks | 600 | 13 |
| Creative Stacked Block | 600 | 14 |
| General | 600 | 12 |

Table 3: **Datasets statistics.** The total number of episodes and corresponding number of expert demonstrations ($\mathcal{D}_E$) per task.

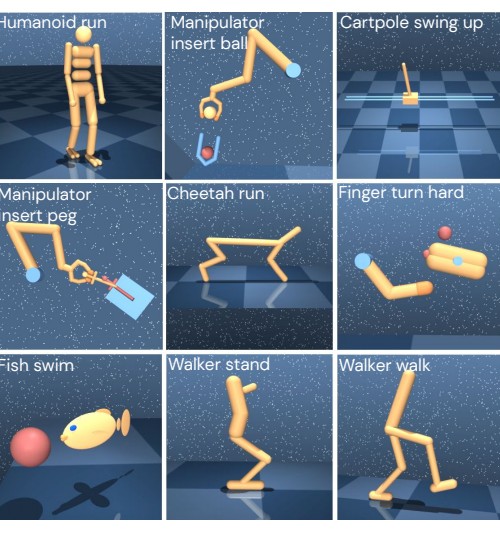

Figure 6: DEEPMIND CONTROL SUITE is a set of popular continuous control environments with tasks of varying difficulties, including locomotion and simple object manipulation.

**CAUSALWORLD** CAUSALWORLD provides a combinatorial family of such tasks with common causal structure and underlying factors (including, e.g., robot and object masses, colors, sizes) (Figure

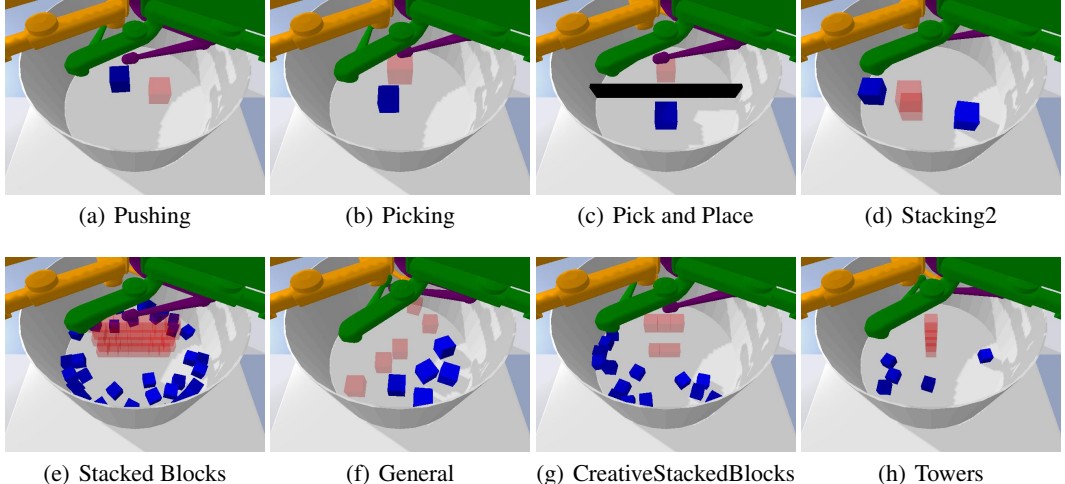

|        (a) Pushing        |        (b) Picking        |      (c) Pick and Place      |        (d) Stacking2        |
| (e) Stacked Blocks | (f) General | (g) CreativeStackedBlocks | (h) Towers |

Figure 7: Example tasks from the task generators provided in the CAUSALWORLD. The goal shape is visualized in opaque red, and the blocks are visualized in blue.

7). We conduct the offline dataset collection process by using various online behavior policies. We collect the mixed dataset by using three kinds of do-interventions (Figure 8) on different environment features. And we divide the offline dataset into $\mathcal{D}_E$ and $\mathcal{D}_U$, similar to the DEEPMIND CONTROL SUITE. The detailed statistics about the dataset are presented in Table 3.

### E.2    Detailed Descriptions of Baselines

- **BC-exp:** Behavioral cloning on expert data $\mathcal{D}_E$. $\mathcal{D}_E$ owns higher quality data but fewer quantities and thus causes serious compounding error problems to the resulting policy.

- **BC-all:** Behavioral cloning on all data $\mathcal{D}_{all}$. BC-all can generalize better than BC-exp due to access to a much larger dataset, but its performance may be negatively impacted by the low-quality data in $\mathcal{D}_{all}$.

- **ORIL** [39]**:** ORIL learns a reward function and uses it to solve an offline RL problem. It suffers from high computational costs and the difficulty of performing offline RL under distributional shifts.

- **BCND** [26]**:** BCND is trained on all data, and it reuses another policy learned by BC as the weight of the original BC objective. Its performance will be worse if the suboptimal data occupies the major part of the offline dataset.

- **LobsDICE** [12]**:** LobsDICE optimizes in the space of state-action stationary distributions and state-transition stationary distributions rather than in the space of policies.

- **DWBC** [36]**:** DWBC is trained on all data. It mainly designs a new IL algorithm, where the discriminator outputs serve as the weights of the BC loss.

## F    Additional Results

### F.1    In-distribution Experiments on CAUSALWORLD

To further show the in-distribution performance, we supplement the experiments on CAUSALWORLD, in which both training and testing are conducted on space **A**. The results are shown in Table 4. In most tasks, our OILCA still achieves the highest average episode return, demonstrating our method's effectiveness across different scenarios. Especially comparing the results in Table 2 and Table 4, we can notice that the advantage of OILCA for out-of-distribution generalization is more obvious. This proves the strong generalization ability of the counterfactual data augmentation module, which makes the offline imitation learning policy more robust to the data distribution shift. This point is especially significant in out-of-distribution scenarios, where the data distribution shifts more intensely.

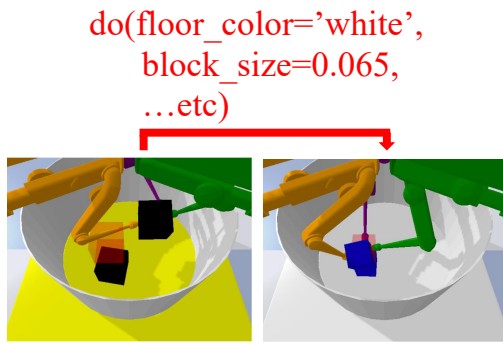

Figure 8: Example of *do*-interventions on exposed variables in CAUSALWORLD.

Table 4: Results for in-distribution performance on CAUSALWORLD. We report the average return of episodes (length varies for different tasks) over five random seeds. All the models are trained on *space* **A** and tested on *space* **A** to show the in-distribution performance [2]. The best results and second best results are **bold** and underlined, respectively.

| Task Name | BC-exp | BC-all | ORIL | BCND | LobsDICE | DWBC | **OILCA** |
|---|---|---|---|---|---|---|---|
| REACHING | 353.98 | 247.60 | 372.39 | 358.36 | 323.43 | 530.96 | **986.19** |
| | ± 11.48 | ± 15.99 | ± 9.58 | ± 15.45 | ± 10.13 | ± 8.70 | ±10.27 |
| PUSHING | 331.32 | 310.62 | 364.37 | 335.87 | 275.38 | 436.22 | **579.55** |
| | ± 6.36 | ± 9.21 | ± 8.36 | ± 9.02 | ± 9.93 | ± 5.39 | ± 12.64 |
| PICKING | 394.63 | 360.28 | 427.39 | 381.45 | 326.97 | 479.05 | **648.34** |
| | ± 12.98 | ± 8.98 | ± 13.69 | ± 8.63 | ± 12.31 | ± 8.57 | ± 8.51 |
| PICK AND PLACE | 453.59 | 355.83 | 348.34 | 376.34 | 287.81 | 448.89 | **588.87** |
| | ± 7.58 | ± 8.47 | ± 11.63 | ± 9.87 | ± 10.06 | ±12.49 | ± 9.29 |
| STACKING2 | 596.14 | 435.12 | 467.11 | 476.33 | 378.3 | 631.75 | **920.18** |
| | ± 15.76 | ± 12.81 | ± 13.19 | ± 5.21 | ± 7.65 | ± 8.54 | ± 7.36 |
| TOWERS | 723.49 | 947.96 | 679.93 | 680.61 | 735.79 | 915.26 | **1263.94** |
| | ± 15.82 | ± 17.56 | ± 8.68 | ± 8.57 | ± 12.23 | ± 17.97 | ± 8.98 |
| STACKED BLOCKS | 1320.97 | 947.96 | 1520.62 | 1247.96 | 958.64 | 2116.51 | **3210.23** |
| | ± 19.83 | ± 25.45 | ± 31.62 | ± 29.14 | ± 26.56 | ± 32.97 | ±43.63 |
| CREATIVE STACKED BLOCKS | 684.52 | 593.41 | 758.04 | 933.88 | 601.18 | 870.29 | **1476.41** |
| | ± 16.69 | ± 26.86 | ± 12.70 | ± 16.57 | ± 19.42 | ± 24.56 | ± 25.94 |
| GENERAL | 626.15 | 691.37 | **1072.05** | 572.70 | 549.89 | 786.44 | 964.32 |
| | ±20.57 | ±17.22 | ±47.26 | ±11.28 | ±15.31 | ±18.52 | ± 17.08 |

## F.2 Combinations with Other Base Offline IL Methods

To validate that the effectiveness of our method is not restricted by the base offline IL methods, we combine the Counterfactual data Augmentation (CA) part with ORIL and BCND, which are represented as ORIL+CA and BCND+CA, respectively. Also, we conduct corresponding experiments on the benchmarks in this paper, and the results are shown in Table 5. From the table, we can observe that the CA module can always help improve policy performance regardless of the base policy choice, which demonstrates its wide applicability. Besides, referring to the results in Table 5, Table 1, and Table 2, we can find that ORIL+CA and BCND+CA outperform all methods without CA's assistance in most tasks, which implies that the simple counterfactual data augmentation may even work better than the complicated learning method designs.

## F.3 Performance of changing the auxiliary variable $c$

To show the influence of the different choices of the auxiliary variable $c$, we conduct additional experiments on the CAUSALWORLD benchmark. Specially, for the change of $c$'s choice, we apply the similar *do*-interventions to more features (*i.e.* block color, block mass) and fewer features. The performance of our OILCA under different intervened features (different choices of $c$) is shown in Table 6. Specially, $C = 1$ represents feature set `stage_friction`, $C = 2$ represents feature set (`stage_friction`, `floor_friction`), $C = 3$ represents feature set (`stage_color`,

Table 5: Results for in-distribution performance on part of tasks in DEEPMIND CONTROL SUITE and out-of-distribution generalization on part of tasks in CAUSALWORLD. We report the average return of episodes (length varies for different tasks) over five random seeds. The training and testing procedures follow those introduced in Section 5. All the results obtained by CA-assisted methods are **bold** to highlight the effect of the counterfactual data augmentation module.

| Benchmark | Task Name | ORIL | ORIL+CA | BCND | BCND+CA | DWBC | OILCA |
|---|---|---|---|---|---|---|---|
| DEEPMIND CONTROL SUITE | CARTPOLE SWINGUP | 221.24 ± 14.49 | **426.79** ± 12.09 | 243.52 ± 11.33 | **452.68** ± 12.86 | 382.55 ± 8.95 | **608.38** ± 35.54 |
| | CHEETAH RUN | 45.08 ±9.88 | **78.44** ± 6.95 | 96.06 ± 16.15 | **158.62** ± 8.85 | 66.87 ± 4.60 | **116.05** ± 14.65 |
| | FINGER TURN HARD | 185.57 ± 26.75 | **227.94** ± 15.47 | 204.67 ± 13.18 | **284.29** ± 12.03 | 243.47 ± 17.12 | **298.73** ± 5.11 |
| | FISH SWIM | 84.90 ± 1.96 | **156.92** ± 8.18 | 153.28 ± 19.29 | **268.56** ± 6.03 | 212.39 ± 7.62 | **290.29** ± 10.07 |
| CAUSAL WORLD | REACHING | 339.40 ± 12.98 | **652.21** ± 7.05 | 228.33 ± 7.14 | **582.44** ±9.07 | 479.92 ± 18.75 | **976.60** ± 20.13 |
| | PUSHING | 283.91 ± 19.72 | **367.46** ± 6.31 | 191.23 ± 12.64 | **320.94** ± 10.37 | 298.09 ± 14.94 | **405.08** ± 24.03 |
| | PICKING | 388.15 ± 19.21 | **458.03** ±13.95 | 221.89 ± 7.68 | **486.32** ±8.03 | 366.26 ± 8.77 | **491.09** ± 6.44 |
| | PICK AND PLACE | 270.75 ± 14.87 | **372.18** ± 10.74 | 259.12 ± 8.01 | **393.59** ±7.81 | 349.66 ± 7.39 | **490.24** ± 11.69 |

Table 6: Results for under different choice of $c$ on the CAUSALWORLD benchmark (out-of-distribution). We report the average return of episodes (length varies for different tasks) over five random seeds.

| Task Name | $C = 1$ | $C = 2$ | $C = 3$ | $C = 4$ | $C = 5$ |
|---|---|---|---|---|---|
| REACHING | 928.62 ± 22.38 | 957.54 ± 18.39 | 976.60 ± 20.13 | 985.25 ± 17.26 | 1037.12 ± 19.15 |
| PUSHING | 389.16 ± 9.43 | 396.52 ± 17.29 | 405.08 ± 24.03 | 426.60 ± 15.37 | 429.42 ± 12.28 |
| PICKING | 462.54 ± 9.08 | 484.21 ± 11.37 | 491.09 ± 6.44 | 522.96 ± 13.27 | 525.20 ± 12.28 |
| PICK AND PLACE | 464.68 ± 10.27 | 486.74 ± 8.52 | 490.24 ± 11.69 | 511.76 ± 9.05 | 523.46 ± 15.42 |
| STACKING2 | 794.81 ± 16.50 | 803.27 ± 13.26 | 831.82 ± 11.78 | 867.43 ± 9.82 | 871.43 ± 18.19 |
| Towers | 972.34 ± 12.36 | 979.23 ± 8.72 | 994.82 ± 5.76 | 1027.16 ± 17.25 | 1029.37 ± 8.06 |
| STACKED BLOCKS | 2317.48 ± 74.32 | 2558.35 ± 42.17 | 2617.71 ± 88.07 | 2682.76 ± 69.25 | 2754.39 ± 82.16 |
| CREATIVE STACKED BLOCKS | 1226.72 ± 62.18 | 1297.20 ± 39.42 | 1348.49 ± 55.05 | 1468.65 ± 27.63 | 1486.51 ± 41.29 |
| GENERAL | 868.62 ± 7.65 | 875.55 ± 19.28 | 891.14 ± 23.12 | 926.19 ± 17.34 | 934.74 ± 16.20 |

`stage_friction, floor_friction`), $C = 4$ represents feature set (`stage_color, stage_friction, floor_friction, block_mass`), $C = 5$ represents feature set (`stage_color, stage_friction, floor_friction, block_color, block_mass`).

From the above Table 6, we can find that our OILCA can achieve a consistent performance improvement over different baselines under different choices of $c$. This demonstrates that the empirical performance of our method is relatively robust to the selection of this variable $c$. In fact, when increasing the number of intervened features (the number of $c$ choices), we can observe our model can achieve better performance. This is because the policy can learn to adapt to more diverse/uncertain environment configurations during the training phase.

## F.4 Influence of the augmented data

In order to prove that the performance will not decay when further improving the $D_E/D_U$, we further increase $D_E/D_U$ (larger than 1) and conduct the experiments with three tasks in DEEPMIND CONTROL SUITE of our method OILCA. Moreover, to show the quality of augmented data, we show the performance gap when increasing expert data proportion using two kinds of augmented data: 1) sampling with the policy in the online environment for more true expert data (Expert Data), 2) our counterfactual data augmentation method OILCA (Augmented Data). The experimental results are shown in Table 7.

Table 7: Results for the Influence of the augmented data with improving the proportion of augmented data and comparison to the true expert data in DEEPMIND CONTROL SUITE Benchmark.

| Task Name | CARTPOLE SWINGUP | | CHEETAH RUN | | CARTPOLE SWINGUP | |
|---|---|---|---|---|---|---|
| **Proportion** | Augmented Data | Expert Data | Augmented Data | Expert Data | Augmented Data | Expert Data |
| 10% | 430.21 ± 13.20 | 441.36 ± 12.01 | 71.85 ± 8.26 | 74.56 ± 3.29 | 261.77 ± 14.68 | 255.62 ± 18.29 |
| 30% | 463.78 ± 21.95 | 472.92 ± 7.62 | 86.44 ± 13.62 | 82.06 ± 9.36 | 269.85 ± 13.39 | 272.18 ± 12.25 |
| 50% | 502.81 ± 20.76 | 520.15 ± 15.43 | 92.60 ± 16.51 | 89.21 ± 12.98 | 276.12 ± 9.82 | 285.48 ± 8.36 |
| 70% | 557.90 ± 16.62 | 562.89 ± 20.47 | 105.57 ± 11.29 | 111.27 ± 11.56 | 283.69 ± 12.71 | 295.83 ± 13.48 |
| 90% | 589.01 ± 38.29 | 593.37 ± 16.81 | 113.12 ± 9.25 | 118.32 ± 15.27 | 288.27 ± 7.09 | 306.26 ± 10.81 |
| 100% | 608.38 ± 35.54 | 621.80 ± 9.26 | 116.05 ± 14.65 | 128.07 ± 8.31 | 298.73 ± 5.11 | 303.51 ± 11.67 |
| 200% | 596.52 ± 28.37 | 634.12 ± 18.29 | 106.39 ± 10.08 | 132.64 ± 14.24 | 303.64 ± 12.91 | 311.70 ± 9.74 |
| 300% | 612.30 ± 41.25 | 635.93 ± 25.15 | 118.51 ± 15.72 | 125.18 ± 8.73 | 301.57 ± 8.30 | 305.42 ± 14.53 |
| 500% | 601.47 ± 27.82 | 627.47 ± 22.86 | 109.96 ± 9.84 | 129.72 ± 12.34 | 289.15 ± 15.27 | 302.15 ± 12.16 |
| 1000% | 605.81 ± 31.63 | 629.94 ± 23.28 | 117.08 ± 7.69 | 124.80 ± 9.46 | 295.48 ± 7.84 | 304.93 ± 11.19 |

From Table 7, we can find that the performance will converge when the proportion is close to 100%, and further improving it indeed will not improve the performance obviously. This can be explained by the results that augmenting too much data can hardly bring additional effective information gain to the learned policy. Moreover, our augmented counterfactual data behaves slightly worse than augmentation with true expert data under most proportions, though achieving obvious improvement over other IL baselines. This shows that the augmented data through our method is high-quality enough.

### F.5  Learning Curves of OILCA

We provide the learning curves of OILCA in Figure 9. In detail, we deploy our trained policy to the online environment at each epoch and then collect 100 episodes for computing the average episode return. From the figures, we can observe that the policies can converge after 200 epochs in most tasks. The fluctuation of the curves mainly comes from the instability of the base offline IL method.

## G  Limitation Analysis

We simply analyze the limitations of this work in this section. In this paper, we only provide the theoretical guarantee to the generalization ability of learned policy from the perspective of the counterfactual samples' number. Actually, why the samples generated by the counterfactual augmentation module are more meaningful and can help the learned policy generalize better than samples obtained by other augmentation methods is also worth exploring theoretically.

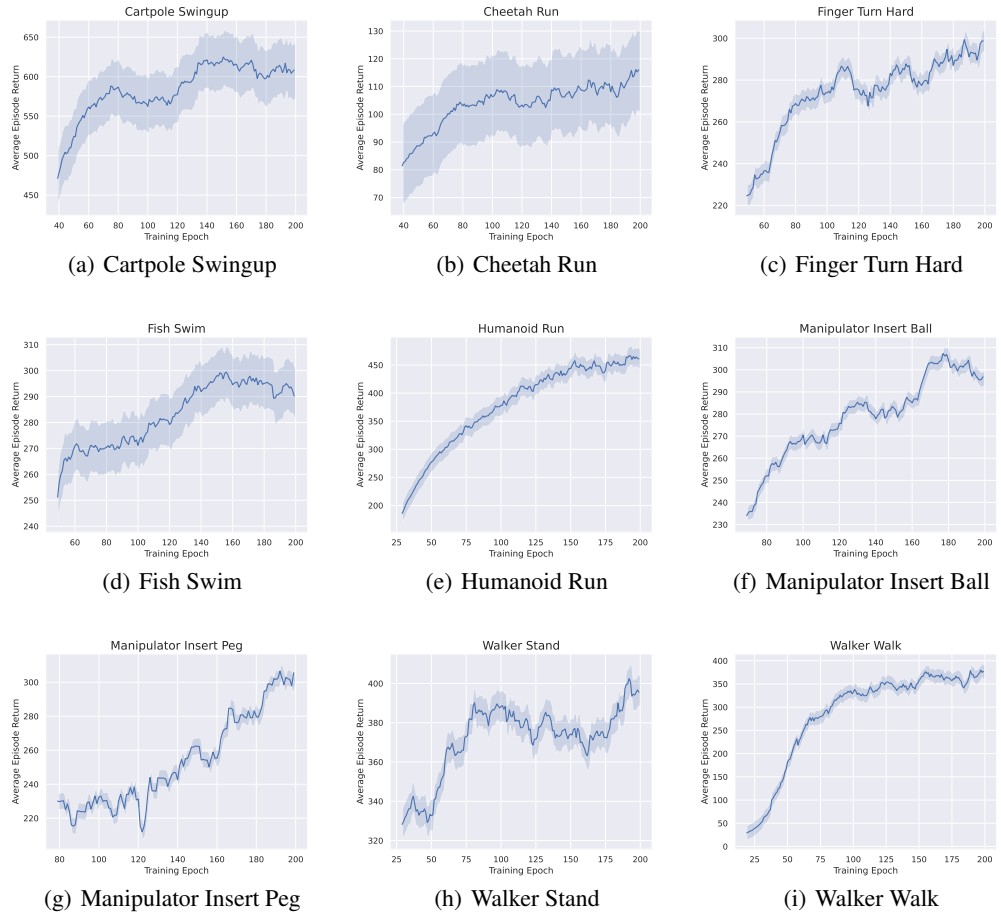

Figure 9: Learning curves of OILCA on 9 tasks of DEEPMIND CONTROL SUITE.