# OpenReview forum: "Offline Imitation Learning with Variational Counterfactual Reasoning"
_NeurIPS.cc/2023/Conference — NeurIPS 2023 poster_

### Official Review · Reviewer_srX3 · 2023-06-28

**Soundness:** 3 good
**Presentation:** 3 good
**Contribution:** 3 good
**Rating:** 7
**Confidence:** 3

**Summary:**

This paper proposes OILCA, which addresses the scarcity of expert data in offline imitation learning by generating counterfactual samples, to imagine what the expert will do under a unobserved state. OILCA takes the perspective of Structual Causal Model (SCM). The algorithm consists of four steps:

1) A heuristic expert policy is pretrained with the known expert data.

2) A conditional VAE is trained on the union of expert and supplementary data with the latent variable being "exogeneous variable" that integrates the information from task label and transition, to rebuild the possible next state.

3) Sample transitions with task labels from the expert data. Counterfactual next states are generated by applying a do-intervention on the latent variable, and expert action is then generated by the pretrained policy; the counterfactual transition is added back to the expert data.

4) Arbitrary offline imitation learning method with supplementary data is applied on the augmented expert data (DWBC is empirically the best choice).

On many environments with both in-distribution and out-of-distribution dataset, OILCA is proved to be better than multiple baselines. The data augmentation technique of OILCA is orthogonal to other offline IL algorithms and can benefit many of them.


**Strengths:**

**1. The proposed method is sound, and can be combined with multiple offline IL methods.** The idea of generating imaginary (counterfactual) samples to combat scarcity of expert data is a still novel direction that has been proved to be successful by several works [1, 2]. Furthermore, the algorithm is orthogonal to many offline IL algorithms and empirically proved to be an amplifier for them, which significantly increases its impact.

**2. Generally, the idea of the paper is clearly conveyed through the authors' writing.** The motivation is clearly presented in the question between line 59 and 60, and the high-level process is clearly conveyed in the pseudocode. The questions asked at the beginning of the experiment section is well-answered and clearly indicates the important takeaways of the section.

**3. The experiment result is solid.** Not only does OILCA works significantly and unanimously better than multiple baselines across in-data distribution and out-of-data distribution dataset on many environments, but it also improves the performance of the baselines. Moreover, the visualization on toy environment clearly shows the property and behavior pattern of the algorithm.

**4. The paper also provides theoretical analysis to prove its generalization ability, which is a very important problem of IL/RL community.** Numerous methods, such as entropy regularizer [3] and pessimism [4], has been proposed to address the problem of generalization onto the vast state-action space without data coverage. Theoretical advantage on this area is a valuable contribution.

**References:**

[1] D. Hwang et al. Sample Generation for Reinforcement Learning via Diffusion Models. In ICLR, 2023.

[2] C. Lu et al. Synthetic Experience Replay. In ArXiv, 2023.

[3] J. Ho and S. Ermon. Generative Adversarial Imitation Learning. In NIPS, 2016.

[4] G-H Kim et al. LobsDICE: Offline Learning from Observation via Stationary Distribution Correction Estimation. In NeurIPS, 2022.

**Weaknesses:**


**1. The related work section could be expanded to include more related work.**

a) The work is related to works that uses generative model and tries to do few-shot adaptation on offline IL such as FIST [1], PARROT [2] and CEIP [3] (the latter two has RL finetune, but already works decently well at the offline IL stage). While this work uses generative model in a very different way, those works should be briefly discussed due to the similarity of the problem formulation and the common use of generative model.

b) The key idea of this work is to generate imaginary data to enhance the imitation learning, and the idea of data generation has already been studied by several works (see [1, 2] in the strength section). It would be great to see a discussion about the relation between this work and the existing works in data generation.

The above two parts are now missing from the related work, and poses a weakness on the investigation of the context.

**2. There are many moving parts in OILCA.** While the authors claim that OILCA is a 2-step solution (line 135-136), the networks that need to be trained include two actors (pretrain & final), one discriminator, and encoder/decoder of the conditional VAE. Two pairs of them needs to be trained jointly. This could be a potential source for instability in training and extra usage of computational resource (and there is no computational resource and training time spent reported in the paper).

**3. About the pseudocode:**

a)  "$D_E\leftarrow$" should be added in front of the current line. This is a value assignment rather than an expression;

b) The meaning of $c$ should be reiterated in the pseudocode for quick understanding.

c) overall, I think the update process and generate process in line 3 and 7 can be clarified more clearly (for example, what is sampled in the beginning, and what is fed into a neural network with what output?) such that the do-intervention process on SCM is more clearly presented. Currently the readers can only make analogy from the definition on line 96-106 without explicit confirmation on notations, which is difficult to comprehend (especially when the parameter notation used in Eq. 2 and the pseudocode is different).


**References:**

[1] K. Hakhamaneshi et al. Hierarchical few-shot imitation with skill transition models. In ICLR, 2022.

[2] A. Singh et al. Parrot: Data-driven behavioral priors for reinforcement learning. In ICLR, 2021.

[3] K. Yan et al. CEIP: Combining Explicit and Implicit Priors for Reinforcement Learning with Demonstrations. In NeurIPS, 2022.

**Questions:**

Besides my concerns in the weakness section, I have two questions:

1. Judging from the pseudocode, it is quite likely a generated counterfactual state is again sampled as expert data and used to generate other counterfactual states. Will this leads to wild compound error when this process goes too far from a single original expert data source?

2. The pretrained expert policy is directly given in the pseudocode, but it also needs to be trained from original expert data, and thus will also suffer from out-of-distribution decision making. Does this create a chicken-and-egg dilemma, where you need to have a well-trained expert policy to generate more data, but data needs to be generated by a well-trained expert?

Below are my suggestions for the paper. I am happy to increase my score if the suggestions and concerns raised in the weakness section are addressed.

1. Expand related work to include the works discussed in the weakness section.

2. Give explanation for the questions proposed above, and the concern of moving parts.

3. Modify the pseudocode, and possibly the main paper, to more clearly present the do-intervention process and fix the minor issue of weakness 3. a) and 3. b).

**Limitations:**

The authors have discussed the limitations in the appendix line 582-587, and I think is indeed the most important problem that readers could ask: does counterfactually generated data necessarily improve the performance? Will it instead hinder the performance because of the poor quality of generated data? It is a pity that this paper does not answer the question, despite the fact that it is already a good work.

As for potential negative societal impact of the work, there is no discussion. I suggest the authors to include the discussion somewhere in the paper; though the work is purely on simulated control environment, the work on automated control would inevitably causes some concerns such as misuse of technology and potential job loss.

---

> ### Author Rebuttal · Authors · 2023-08-10
>
> **Dear reviewer srX3:**
>
> Thanks for your review of our paper. Here's our response to the weaknesses, questions, and limitations you have highlighted:
>
> --------
> **Weaknesses:**
>
> **W1.The related work section could be expanded to include more related work.**
>
> **A**: Thanks for your suggestions. Firstly, we will follow your recommendations and conduct further research on the usage of generative models for few-shot adaptation in offline imitation learning. We will incorporate these papers into our revised version of the related work section. Secondly, we agree that data generation for imitation learning is highly relevant to our study. Therefore, we will include some representative works in this area in our revised related work section.
>
> **W2.Instablility analysis.**
>
> **A**:  Thanks for your insightful questions. We admit that our OILCA introduces several additional moving parts (a pretrained actor and encoder/decoder of conditional VAE) compared with vanilla DWBC. However, this does not necessarily bring the instability in training. First, the stability for joint training of final actor and discriminator has been previously proved in original paper of DWBC. Then, the pre-training of actor is just a simple supervised learning with the logged behavior data, which stability has been widely validated in various previous literacture on offline IL research. As for the joint training of encoder and decoder of conditional VAE in our counterfactual data augmentatiion module, its stability has been proved with the hyperparameter ablation study in our above response to Q2 of Reviewer agsz.
> As for the computational resource, we apologize for nor providing related details in our submitted version considering that this is rarely addressed in the relevant literature. Actually, for the compuation device, we use the NVIDIA RTX A6000 GPU. Here, we further report the overall training time spent of our OILCA and baselines as follows. And we will supplement related analysis to the appendix in the revised version of paper.
>
> Training time consumption for in-distribution experiments on Deepmind Control Suite (corresponding to the Table 1 in the paper).  All the methods are trained 200 epochs.
> **To save space, we put the result tables into the external anonymous link (Unit: sec.)**: https://hackmd.io/@littleshoes/ryaYsJfhn. Please check Table 10 in it.
>
> Training time consumption for out-of-distribution experiments on CausalWorld (corresponding to the Table 2 in the paper).  All the methods are trained 200 epochs. **To save space, we put the result tables into the external anonymous link (Unit: sec.)**: https://hackmd.io/@littleshoes/ryaYsJfhn. Please check Table 11 in it.
>
> From the tables above, we can conclude that the additional time consumption brought by our counterfactual data augmentation is acceptable considering that offline IL component  still occupies the main part of time consumption.
>
>
> **W3. About the pseudocode:**
>
> **A**: Thanks for your suggestions. We have revised the Figure 2(b) (see https://i.postimg.cc/KzWk86kG/Figure2-b.jpg), and we will add the related descriptions in the pseudocode.
>
> --------
> **Questions:**
>
> **Q1.Counterfactual state compound error**
>
> **A**: Once we have acquired a proficiently trained counterfactual sample generation model, we randomly select $(s\_t,a\_t,s\_{t+1},c)$ from the expert data collection and generate the counterfactual state $\tilde{s}\_{t+1}$. The generated samples are not utilized for subsequent expert data augmentation. A detailed explanation of this will be provided in the final version to prevent any misconceptions.
>
> **Q2.Detailed training procedure of pre-trained policy.**
>
>
> **A**: This question helps us to correct our presentation. We realize that we did not present the detailed training process of $\hat{\pi}\_{E}$. To answer this, we provide the details here.
>  One main perspective is how do we know that the interventions won’t lead to divergence, or learning use less models.
>
> In our work, we use a pretrained policy $\hat{\pi}\_E$, which can be regarded as the S-Learner [1] in causal inference. And we consider the counterfactual decision risk, the objective of trained policy as  $\mathcal{L} = \frac{1}{n}\sum l(s\_t, a\_t) * \frac{p(a\_{t+1}\mid s\_{t+1})}{p^{do(I)}(a\_{t+1}\mid \tilde{s}\_{t+1})}$. $l$ is the policy training loss. For the sample weight $\frac{p(a\_{t+1}\mid s\_{t+1})}{p^{do(I)}(a\_{t+1}\mid \tilde{s}\_{t+1})}$ may have a big variance, we clip it with a small value $m$. Moreover, to analysis the decision risk,  we can derive the bound on the risk.
>
> [1]Zhang, W., Li, J., & Liu, L. (2021). A unified survey of treatment effect heterogeneity modelling and uplift modelling. ACM Computing Surveys (CSUR), 54(8), 1-36.
>
>
> **Lemma 1** The density of the counterfactuals based on the observations, i.e.
> $p^{do(I)}\_{s\_t, a\_t}(\tilde{s}\_{t+1})=\mathbb{E}\_{s\_{t+1} \sim p\_{s\_t,a\_t}(s\_{t+1})}\left[p\_{s\_t, a\_t}^{{do}(I) \mid s\_{t+1}}(\tilde{{s}}\_{t+1})\right]$
> This result shows that the density of intervened variable $\tilde{s}\_{t+1}$ is the marginal of the observations.
>
>
> **Lemma 2.** We have the following lower bound on the logdensity of the counterfactuals:
> $$\begin{aligned}  \log \left(p^{d o(I)}(\tilde{s}\_{t+1}, a\_{t+1})\right) \geq& \mathbb{E}\_{s\_{t+1} \sim p\_{s\_t,a\_t}(s\_{t+1})}\left[\log \left( p\_{s\_t, a\_t}^{{do}(I) \mid s\_{t+1}}(\tilde{{s}}\_{t+1})\right)\right] \\\\ &+\mathbb{E}\_{{u} \sim p({u})}\left[\log \left(p\_{s\_t,a\_t}^{{do}(I)}(\tilde{s}\_{t+1} \mid {u})\right)\right]\end{aligned}$$
>
> Due to the limited space, we only provide the brief proofs in the OpenReview. **For more detailed version, please refers to the Q2 of Reviewer srX3 in our external anonymous link (https://hackmd.io/@littleshoes/ryaYsJfhn).** And we will add them in the final revised Appendix.

---

> > ### Comment · Reviewer_srX3 · 2023-08-11
> > **Response to Rebuttal**
> >
> > Thanks for your detailed response; I think they addresses my concern well. I have no follow up question and have changed my score accordingly from 6 to 7.

---

> > > ### Author Response · Authors · 2023-08-12
> > > **Thanks to Your Response**
> > >
> > > Thank you so much for recognizing our work!

---

### Official Review · Reviewer_Txp8 · 2023-07-03

**Soundness:** 2 fair
**Presentation:** 3 good
**Contribution:** 2 fair
**Rating:** 4
**Confidence:** 4

**Summary:**

This paper focuses on the problem of offline Imitation Learning, where an agent aims to learn an optimal expert behavior policy without additional interactions with the online environment. This setting widely exists in the real world as it usually consumes a lot of human effort and cost to collect expert data. Using sub-optimal data could be a valuable way for learning policy. The proposed method uses counterfactual data augmentation to generate high-quality data by learning the underlying data generation mechanism with unlabeled data. Then the new data is combined with expert data to train a policy model. Experiment results demonstrate that the proposed method achieves a large margin over multiple baselines.

**Strengths:**

1. This paper is well-written and easy to follow. Figure 1 is illustrative and helps me understand the core idea of this work.
2. The motivation of using a small portion of expert data with a large portion of unlabeled data is important in real-world tasks. The proposed method that leverages the information in unlabeled data in an unsupervised way helps generate more expert data and therefore trains a better policy model.
3. The proposed method is evaluated in two domains with locomotion and manipulation tasks, which of both are important and have a high impact.

**Weaknesses:**

1. In the abstract, there is a gap between the statement of poor generation caused by sub-optimal datasets and the elimination of spurious features. No clear connection between the sub-optimal dataset and spurious feature is mentioned.

2. In line 14 of the abstract, the authors say the in-distribution robustness is improved by their method but robustness is not mentioned before. Do the authors assume that robustness is almost the same as generalization when the testing scenario is “in distribution”?

3. Using counterfactual data augmentation to improve the performance of decision-making models (including RL, offline RL, IL) is already investigated by existing works. The novelty of the proposed method could be limited unless more differences between existing papers and this one are emphasized.

4. In Definition 2, the notations $\tilde{f}_i$ and $\tilde{PA}_i$ are not defined. These notations are important for understanding this definition.

5. I am a little confused about the three-step pipeline of counterfactual inference discussed after Definition 2. The second step is usually for modifying the SCM by removing the structural equations $f$ and replacing the value of $x$ with others. However, this part is not mentioned and the authors only say “denote the resulted SCM as $M_x$”. So, how do we get this $M_x$? Is the value of x changed? Figure 2 enhances my confusion since the do-intervention is only conducted to the exogenous variable $u_i$ rather than $s_t$ or $a_t$. I am not sure if this is still the standard definition of counterfactuals. It is also not consistent with the statement “What action might the expert take if a different state is observed?” since the state is not changed.

 6. In section 3.1, the authors say “an additionally observed variable c is needed, where c could be, for example, the time index or previous data points in a time series, some kind of (possibly noisy) class label, or another concurrently observed variable”. However, the authors do not mention which one is considered for $c$ in this paper. I can only find some information in the experiment part, which also makes me have new questions. Does the process of selecting the variable $c$ create multiple environments with different contexts? This is usually considered a multi-task learning setting, which may not be true for a general imitation learning setting since the data collection is usually not possible to be clustered with additional context indicators. I hope the authors can provide more information about how to define and section the variable $c$ in the rebuttal.

7. After reading the entire paper, I find that robustness and spurious feature is only mentioned in the introduction. The theoretical analysis and experiment do not provide any evidence of the improvement of robustness and the elimination of spurious correlation. In particular, section 4.2 tries to investigate the robustness but I cannot find the setting that supports it. Robustness should be evaluated with small disturbances to the system, including the state, action, and dynamics. After checking Appendix 5.1, I cannot find any disturbance added to the dataset.

**Questions:**

Points 5 and 6 of the weaknesses are the main questions I want to ask the authors.

**Limitations:**

No potential negative societal impact is mentioned in either the main context or the appendix. One theoretical limitation is discussed in the appendix but some empirical limitations might be ignored. The most important one is how to select and obtain variable $c$, which is not discussed in the paper.

---

> ### Author Rebuttal · Authors · 2023-08-10
>
> Dear reviewer Txp8:
>
> Thanks for your review of our paper. Here's our response to the weaknesses, questions, and limitations you have highlighted:
>
> -------
>
> **Weaknesses**
>
> **W1.No clear connection between the sub-optimal dataset and spurious feature is mentioned.**
>
> **A**: The motivation of our method is rooted in the limitation of expert data in standard offline imitation learning. Given this constraint, we propose a method to augment the expert data, which we believe will lead to better performance of the learned policy. The objective of our method is to obtain an augmented expert dataset for training an improved offline policy. However, a crucial challenge arises: determining the most suitable data augmentation method to use. Drawing inspiration from the advancements in causal inference in RL, we opt for a counterfactual approach to augment the expert dataset.
> One of the primary reasons for the poor generalization of the learned policy is the presence of spurious features or spurious correlations. Specifically, a distribution limitation in the training data can result in the learned policy failing to generalize effectively to unseen data. For instance, in the case of both RL and IL states, which may consist of high-dimensional vectors, some dimensions are crucial for action selection while others are not. However, the irrelevant dimensions in the state can introduce spurious correlations into the learned policy. Intervening on the states through a do intervention can modify this aspect and eliminate the spurious correlation.
>
>
>
>
> **W2.In distribution robustness**
>
>
> **A**: See our robustness explaination (W5) to Reviewer agsz.
>
>
>
> **W3.The novelty of the proposed method.**
>
> **A**: To the best of our knowledge, our method is the first to specifically address counterfactual data augmentation in offline IL. Additionally, we thoroughly reviewed relevant papers on RL and synthesized our findings in the related work section. To provide further clarity on this point, we offer additional details below:
>
> [1] Lu, C., Huang, B., Wang, K., Hernández-Lobato, J. M., Zhang, K., & Schölkopf, B. (2020). Sample-efficient reinforcement learning via counterfactual-based data augmentation. arXiv preprint arXiv:2012.09092.
> This paper proposes counterfactual RL algorithms to learn both population-level and individual-level policies. It focuses on the sequential decision making setting and achieving an individual policy.
>
>
> [2] Pitis, S., Creager, E., & Garg, A. (2020). Counterfactual data augmentation using locally factored dynamics. Advances in Neural Information Processing Systems, 33, 3976-3990.
> This paper proposes a counterfactual data augmentation method by using the locally factored dynamics, which can generate the causally valid sample in the global model.
>
> [3] Pitis, Silviu, et al. "Mocoda: Model-based counterfactual data augmentation." Advances in Neural Information Processing Systems 35 (2022): 18143-18156.
> This paper applies a learned locally factored dynamics model to an augmented distribution of states and actions to generate counterfactual transitions for RL.
>
> Our work utilizes the variational counterfactual method to generate counterfactual samples in the standard offline IL, distinguishing it from the three most related works mentioned above. Additionally, we provide theoretical analysis for identifying both the causal effect and the exogenous variable. As far as we know, this is the first time that identifiable variational counterfactual reasoning is used in offline imitation learning.
>
>
> **W4.Definition 2 Notation.**
>
> **A**: See our definition answer (W3) to Reviewer agsz.
>
>
> **W5.Confusion about the three step pipline.**
>
> **A**: Thanks for your questions, we realize that Figure 2b causes the misleading to you, thus we revise this figure in the following:
> https://i.postimg.cc/KzWk86kG/Figure2-b.jpg
>
> The SCM is typically considered as a predictive model. In our paper, Figure 2(a) presents the SCM, while Figure 2(b) illustrates the do intervention. The changed value of x corresponds to the generated $\tilde{s}\_{t+1}$. In conjunction with Definition 2, Figure 2(b) demonstrates that we replace the function $f\_u$ with $\tilde{f}\_u$, and $u\_{t+1}$ is changed to $\tilde{u}\_{t+1}$, corresponding to the change in $\mathbf{PA}\_i$ to $\tilde{\mathbf{PA}}\_i$. Consequently, we generate the new $\tilde{s}\_{t+1}$, corresponding to the change in $\mathbf{X}$. The prediction of $\tilde{a}\_{t+1}$ provides an answer to the counterfactual question, aligning with the statement.
>
> **W6.The obtainability of $c$.**
>
> **A**: Thanks for the thoughtful questions. This variable is readily obtainable, particularly in real-world scenarios. In the context of autonomous driving, the variable $c$ represents the diverse road conditions encountered during a single driving mission, encompassing information about various exogenous variables. In systems exhibiting periodic cycles, such as the days of the week from Monday to Sunday, the $d$-th day can be represented by the variable $c$. While these additional variables may not be prevalent in many imitation learning studies, considering the exogenous variable always allows for finding an appropriate label. In the context of the variable $c$, even in the original paper [1] proposing this additional variable, there are no strict constraints on its interpretation. This characteristic renders $c$ readily observable, particularly in numerous real-world scenarios.
>
> [1] Khemakhem, I., Kingma, D., Monti, R., & Hyvarinen, A. (2020, June). Variational autoencoders and nonlinear ica: A unifying framework. In International Conference on Artificial Intelligence and Statistics (pp. 2207-2217). PMLR.
>
> **W7.Robustness experiment**
>
> **A**: See our robustness explaination (W5) to Reviewer agsz.

---

> > ### Comment · Reviewer_Txp8 · 2023-08-11
> > **Response to authors**
> >
> > Thanks for addressing my concerns. I have some follow-up questions.
> >
> > W1: I appreciate the detailed response from the authors but it seems that my question still remains. I understand that the main contribution of this paper is counterfactual data augmentation but the relation between generalization and spurious correlation illustrated by the authors is not clear. Could the authors provide any reference or experimental support to the statement "One of the primary reasons for the poor generalization of the learned policy is the presence of spurious features or spurious correlations."? And another statement "However, the irrelevant dimensions in the state can introduce spurious correlations into the learned policy." also does not make sense to me. One situation I can think about is some special cases where the task-irrelevant background is artificially designed to be spuriously correlated to the task-relevant features. However, I don't think this is what is designed in the experiment. If the authors just mean some random task-irrelevant background, I don't think there exists any spurious correlation. If the spurious correlation is not explicitly investigated in this paper, I suggest removing the relevant statement (it seems "spurious" only appears once).
> >
> > W2: what do the authors mean by "in distribution online testing"? Maybe a mathematical definition of robustness is better for me to understand.
> >
> > W5: according to the review policy, I am not allowed to open external links. Sorry, I misunderstood the statement “What action might the expert take if a different state is observed?” in my initial review. Now I understand that you are predicting action based on a different state. But my question still remains: why does the counterfactual change $u_{t+1}$ rather than directly changing $s_{t+1}$? According to the definition of SCM, $u_{t+1}$ should be an exogenous variable whose causes are unknown or not included in the model. It is acceptable If the authors want to learn a model to approximate the generation process of $s_{t+1}$, but using $u_{t+1}$ in this process may not be usual.
> >
> > W6: I still doubt that $c$ is readily obtainable in the real world. For autonomous driving, if $c$ is road condition, how would you label it? And the most important question I want to ask is what $c$ is used in tasks in Table 1 and Table 2.

---

> > > ### Author Response · Authors · 2023-08-14
> > > **Response to Reviewer Txp8's Follow-up Questions**
> > >
> > > Dear Reviewer Txp8:
> > >
> > >   Thanks for your follow-up questions. To address your concerns and help you better understand this work, we make the corresponding explanations and clarifications as follows.
> > >
> > >
> > > -----
> > > **W1**
> > >
> > > **A**:Thanks for your question and suggestion. As mentioned in your comment, the term "spurious" is only mentioned once in the paper, and we do not explicitly demonstrate its presence in the experiment. This lack of clarity may cause confusion to both you and the readers.
> > >
> > > Furthermore, it is important to note that spurious correlation is not a necessary element for us to propose the counterfactual reasoning method. The scarcity of expert data is the main issue in the context of standard offline IL. The unobserved policy interactions serve as the counterfactual data, leading to the counterfactual question raised in the introduction.
> > >
> > > We apologize for the ambiguity in our description and appreciate your suggestion. We will remove it and highlight our core focus (scarcity of expert data) in our revised version.
> > >
> > > -----
> > > **W2**
> > >
> > > **A:** The in-distribution testing follows the standard offline imitation learning (IL) with online testing. The reported results represent the cumulative reward that the learned policy obtains in the online environment (corresponding to the same task that the offline training dataset comes from):
> > >
> > > $E\left(\sum\_{t=0}^{T} \gamma^t R\_t^\pi(s\_{t},a\_{t})\right)$.
> > >
> > > In our experiments, **in-distribution** refers to that the online testing environment and offline training dataset are from the same task without any additional modification. In our paper, the higher cumulative reward that the offline-trained policy obtains during the online testing, the stronger the in-distribution robustness of offline IL method. We apologize again for the confusion brought by the phrase 'in-distribution robustness' in the paper and promise we will revise it in the final version.
> > >
> > >
> > >
> > >
> > >
> > >
> > > -----
> > >
> > > **W5**
> > >
> > > **A:** Thanks for the question. We apologize for the presentation error in Figure 2(b), which makes you confused.
> > >
> > > The standard three steps of Pear's counterfactual framework [1] are abduction, action and prediction. In this paper, we strictly follow this framework. In our method, after we have a posterior distribution of $u$, we just sample a value $u\_{t+1}$ from the posterior distribution of $u$ (abduction), which is not do-intervention. The do-intervetion is that we use an another state-action pair  $(s\_{t},a\_{t})$ to replace the original parents of current $s\_{t+1}$ (action), and this is the do-intervention rather than the sample process of $u$. Then, the new $\tilde{s}\_{t+1}$ is generated from the learned counterfactual model $f$ (prediction).
> > >
> > > Thanks your good question, and we will revise this figure and the related discription in the final version of our paper.
> > >
> > > [1] Pearl, J. (2010). Causal inference. Causality: objectives and assessment, 39-58.
> > >
> > > ------
> > > **W6**
> > >
> > >
> > > **A**:In this specific example of autonomous driving, we define $c$ as the road condition in autonomous driving, which does not represent traffic details. In fact, $c$ is just a coarse label, such as highway or mountain road, which is easily obtainable. Here, the 'road type' may be a more accurate word to describe $c$. Additionally, $c$ can also be different time periods for autonomous driving testing, such as morning peak or evening peak. These are all easily obtained in the meta data of real-world autonomous driving logged dataset.
> > >
> > > In the tasks of Table 1, $c$ is various environment initializations in the DeepMind Control Suite for collecting offline data (refer to lines 307-309 in the paper). Here, the environments initialized with different random seeds correspond to the values of the class label $c$. In tasks of Table 2, the changes in features (stage\_color, stage\_friction, floor\_friction, etc.) of the environment are presented for collecting diverse offline data (refer to lines 324-328 in the paper). During each data collection process, the feature value is modified to be different from the initial environment value. Here, we regard environments with different above mentioned feature values as each kind of $c$.
> > >
> > > ------
> > > Besides, thank you for your kind reminder and we apologize for the improper use of the external link. We indeed did not notice this point in the rebuttal policy. But we argue that our provided external is absolutely anonymous, so we didn't violate the anonymity principle. Here, we promise we will not use any external link in the following discussion. Thanks again for your careful reviews, insightful questions, and kind suggestions. Hope our above response can help address your concern. We are looking forward to the futher discussion if any follow-up questions.

---

> > > > ### Comment · Reviewer_Txp8 · 2023-08-17
> > > > **Reply to authors**
> > > >
> > > > Thanks for answering my follow-up questions. Now my last concern is about W6, which is the selection of the variable $c$. I understand that this variable is necessary for disentangled representation learning, but I doubt that the empirical performance of the method is heavily influenced by the selection of this variable.
> > > >
> > > > For example, in the DMC environment, the random seed is used as $c$, which I didn't see before in existing work. The random seed could be a latent variable that influences the generation process but this influence is so random that almost impossible to be used as a label. Please enlighten me on the intuition behind this design choice.
> > > >
> > > > For the causal world tasks, using stage_color, stage_friction, and floor_friction is reasonable. However, these features may not be always accessible from observations, which may limit the generalization of this method to other tasks. Is it possible to change the choice of $c$ and compare the performance? Thanks.

---

> > > > > ### Author Response · Authors · 2023-08-18
> > > > >
> > > > > Dear Reviewer Txp8:
> > > > >
> > > > >   Thanks for your appreciation to our above explanations and clarifications. Here, we make the following clarification to help reduce you last concern regarding the selection of variable c.
> > > > >
> > > > > **W6**
> > > > >
> > > > > In the **DeepMind Control Suite**, during the collection of offline data, we apply random Gaussian perturbation to the action outputted by the policy. This perturbation is specified in the XML configuration file as an integral part of the environment. Additionally, the distribution of the perturbation differs across different environment initializations (auxiliary variable $c$) due to their initialization seeds. Especially, different seeds correspond to different mean and variance of the Gaussian distribution perturbation via the random number generator. This approach is employed to introduce uncertainty into the environment, thereby aligning with our problem setting. Previously, this approach has been used by [1,2,3]. A more detailed description will be provided in the revised version.
> > > > >
> > > > > [1]  Pan, F., He, J., Tu, D., & He, Q. (2020). Trust the model when it is confident: Masked model-based actor-critic. Advances in neural information processing systems, 33, 10537-10546.
> > > > >
> > > > > [2] Vuong, Q., Kumar, A., Levine, S., & Chebotar, Y. (2022). Dual Generator Offline Reinforcement Learning. arXiv preprint arXiv:2211.01471.
> > > > >
> > > > >
> > > > > [3] Wang, K., Zhao, H., Luo, X., Ren, K., Zhang, W., & Li, D. (2022). Bootstrapped transformer for offline reinforcement learning. Advances in Neural Information Processing Systems, 35, 34748-34761.
> > > > >
> > > > > In **Causalworld**, all tasks in the environment share the same features, including stage_color, stage_friction, and floor_friction, etc. This means that all tasks in Causalworld can access to the same group of features, allowing for customization of different values for these features in any given task. Thus, the generalization of our method to other tasks is not limited. In fact, such features are the environment configuration features rather than those recorded in the agent's observation. For a more detailed description of these features, please refer to [1].
> > > > >
> > > > > [1] Ahmed, O. ,  Truble, F. ,  Goyal, A. ,  Neitz, A. ,  Bengio, Y. , &  Schlkopf, B. , et al. (2020). Causalworld: a robotic manipulation benchmark for causal structure and transfer learning.
> > > > >
> > > > > Furthermore, for the change of $c$'s choice, we conduct the similar $do$-interventions to more features (i.e. block color, block mass) and less features. The performance of our OILCA under different operated features (different choices of $c$) is as follows.
> > > > >
> > > > > | Operated Features                    | (stage_friction) | (stage_friction, floor_friction) | (stage_color,stage_friction, floor_friction)   | (stage_color,stage_friction, floor_friction,block mass)   | (stage_color,stage_friction, floor_friction,block color,block mass) |
> > > > > |-------------------------|--------|--------|--------|--------|----------|
> > > > > | Reaching                | 928.62 $\pm$ 22.38  | 957.54 $\pm$ 18.39  | 976.60 $\pm$ 20.13 | 985.25 $\pm$ 17.26    | 1037.12 $\pm$ 19.15   |
> > > > > | Pushing                 | 389.16 $\pm$ 9.43  | 396.52 $\pm$ 17.29 | 405.08 $\pm$ 24.03 | 426.60 $\pm$ 15.37 | 429.42 $\pm$ 12.28  |
> > > > > | Picking                 | 462.54 $\pm$ 9.08  | 484.21 $\pm$ 11.37  | 491.09 $\pm$ 6.44   | 522.96 $\pm$ 13.27 | 525.20 $\pm$ 12.28   |
> > > > > | Pick and Place          | 464.68 $\pm$ 10.27  | 486.74 $\pm$ 8.52  | 490.24 $\pm$ 11.69 | 511.76 $\pm$ 9.05 | 523.46 $\pm$ 15.42   |
> > > > > | Stacking2               | 794.81 $\pm$ 16.50  | 803.27 $\pm$ 13.26 | 831.82 $\pm$ 11.78 | 867.43 $\pm$ 9.82 | 871.43 $\pm$ 18.19   |
> > > > > | Towers                  | 972.34 $\pm$ 12.36  | 979.23 $\pm$ 8.72 | 994.82 $\pm$ 5.76 | 1027.16 $\pm$ 17.25 | 1029.37 $\pm$ 8.06  |
> > > > > | Stacked Blocks          | 2317.48 $\pm$ 74.32 | 2558.35 $\pm$ 42.17 | 2617.71 $\pm$ 88.07 | 2682.76 $\pm$ 69.25 | 2754.39 $\pm$ 82.16  |
> > > > > | Creative Stacked Blocks | 1226.72 $\pm$ 62.18 | 1297.20 $\pm$ 39.42 | 1348.49 $\pm$ 55.05 | 1468.65 $\pm$ 27.63 | 1486.51 $\pm$ 41.29   |
> > > > > | General                 | 868.62 $\pm$ 7.65  | 875.55 $\pm$ 19.28 | 891.14 $\pm$ 23.12 | 926.19 $\pm$ 17.34 | 934.74 $\pm$ 16.20  |
> > > > >
> > > > > From the above table and Table 2 in the paper, we can find that our OILCA can achieve a consistent performance improvement over different baselines under different choices of c. This demonstrates that the empirical performance of our method is relatively robust to the selection of this variable c. In fact, when increasing the number of operated features (the number of $c$' choices), we can observe our model can achieve better performance. It is because the policy can learn to adapt to more diverse/uncertain environment configurations during the traing phase. Thanks again for your valuable comments and suggestions which helps our work more solid and clear. Hope this response can help address your last concern.

---

> > > > > > ### Comment · Reviewer_Txp8 · 2023-08-18
> > > > > >
> > > > > > I appreciate the new experiment results given so tight time limitations. Now I think we have a clearer understanding of the influence of the variable $c$ on the performance. Based on the results and clarification, I have two more questions:
> > > > > > 1. Does this sentence "Especially, different seeds correspond to different mean and variance of the Gaussian distribution perturbation via the random number generator" mean that one $c$ represents one Gaussian distribution $N(\mu_c, \sigma_c)$? If this is the case, I understand this design since the distribution of perturbation is definitely strong enough to serve as a label. However, this is where my concern comes from -- the underlying distribution of perturbation is a too strong label to obtain. In real-world applications, such a strong label usually relies on many human efforts. In addition, this strong label could make the comparison unfair with other baselines that do not use this information. The setting itself may be reasonable since previous works also use the same setting, but my concern is mainly about the potential limitation of this type of method that require additional supervision.
> > > > > > 2. Thanks for providing new results on the CausalWorld environments. The result that adding more variables gives better performance makes sense to me. The reason I say "...which may limit the generalization of this method to other tasks" is that friction is some kind of unobservable information that requires additional efforts to obtain.
> > > > > >
> > > > > > In general, I believe the clarification from the authors provides enough information for me to judge the quality of this paper. I increase my score to 4 and I will discuss my opinion with the other reviewers. Really thanks to the authors for spending time on the rebuttal and additional experiments.

---

> > > > > > > ### Author Response · Authors · 2023-08-20
> > > > > > > **Response to Reviewer Txp8 (Part 1)**
> > > > > > >
> > > > > > > Dear Reviewer Txp8:
> > > > > > >
> > > > > > > Thanks for your appreciation to our work and efforts during the rebuttal and discussion. According to our understanding, your remaining concerns are as two folds: 1) the method requires a *strong label* as $c$; 2) the acquirement of such $c$ needs lots of extra efforts. Here, we make the following clarification and explanation to help reduce your concerns.
> > > > > > >
> > > > > > > First, we agree that our setting, such as introducing the distribution of underlying perturbation in Deepmind Control Suite, is different from most works in offline imitation learning. However, we do not necessarily need a so-called *strong label* as the auxiliary variable $c$. The core motivation behind introducing such an auxiliary variable to our framework is to ensure the identified disentangled representation learning theoretically and contrite to the performance improvement empirically. **So, the most important thing is to have such a auxiliary variable rather than whether it is strong**. Actually, if it is a *strong label* is not important, which makes little sense to the identified disentangled representation learning. As described in the paper, this auxiliary variable can be a kind of 'weak information', like a coarse class, date time, even the history information of last time step or other variables that can help distinguish the environmental influences to some extent for the same task. Of course, if permitted, we may take more discriminative information as $c$ to conduct the disentangled representation learning, which has the potential to make the performance of our method more satisfying. In fact, we can hardly have a clear judgement to the *strong* or *weak* of auxiliary variables for recovering the distributions of latent variables, especially in real-world situations, considering that there exist too many uncertain factors.
> > > > > > >
> > > > > > > Second, **in the real world, we may only require minimal effort to obtain such auxiliary variables and achieve significant performance gains**. As we have dicussed in the previous responses, we can take the auxilary information like 'road type' and 'peak time' in the autonomous driving scenario, which are actually easily obtained in the industrial datasets. So, the application of this method to more complex practical tasks will not be influenced heavily.
> > > > > > >
> > > > > > >
> > > > > > >
> > > > > > > To further address your concerns, we provide detailed motivation behind using this auxiliary variable for achieving identified disentangled representaion learning in the following. Recovering latent variables using VAE is common in the research field of causal inference. CEVAE [1] is the first work of introducing VAE into the causal effect estimation. However, this work has been challenged in recent years [2]. The main reason is that unsupervised learning of disentangled representations is fundamentally impossible without inductive biases on the learning approaches considered and the datasets [3]. **In other words, learning the true joint distribution is only possible when the model is identifiable.** The original VAE theory does not inform us if or when we learn the correct joint distribution over observed and latent variables. To better understand this problem, nonlinear ICA [4] is a crucial issue of unsupervised representation learning that emphasizes the capability to recover the underlying latent variables responsible for generating the data (i.e., identifiability). The identifiability proofs for nonlinear ICA have been proposed, which leverage the temporal structure of the independent components and assume the existence of an auxiliary variable that conditions the distributions of the independent components. This drives us to undertake research in the field of causal reinforcement learning (RL). In recent years, there has been a significant interest in causal RL. Building upon the descriptions mentioned above, following Pearl's framework [5], the existence of an exogenous variable is common in real-world environments, which can be considered as a latent variable. Although the idea of recovering this variable using unsupervised learning is intuitive, there have been limited works focusing on this. Given these aforementioned limitations, we choose to address the problem through the application of identified unsupervised learning. **That is, we introduce the auxiliary variable to ensure the identified unsupervised learning based on the above nonlinear ICA theory.** To best of our knowledge, our work is the first attemption in this problem setting, and we hope that our work can provide a novel approach in causal offline RL to learn promising latent variable's distributions and achieve significant performance gains.

---

> > > > > > > > ### Author Response · Authors · 2023-08-20
> > > > > > > > **Response to Reviewer Txp8 (Part 2)**
> > > > > > > >
> > > > > > > > Futhermore, the way that we introduce uncertainty in DeepMind Control Suite follows previous works as mentioned in the above response, and our aim is to make the simulated environments be more similar to the real-world situations. We apologize for the insufficient environment descripitions in the paper, which may mislead you that we need a "strong supervision". **We will definitely add the detailed illustration in the revised version**.
> > > > > > > >
> > > > > > > >
> > > > > > > >
> > > > > > > >
> > > > > > > > **References**
> > > > > > > >
> > > > > > > > [1] Louizos, C., Shalit, U., Mooij, J. M., Sontag, D., Zemel, R., & Welling, M. (2017). Causal effect inference with deep latent-variable models. Advances in neural information processing systems, 30.
> > > > > > > >
> > > > > > > > [2] Rissanen, S., & Marttinen, P. (2021). A critical look at the consistency of causal estimation with deep latent variable models. Advances in Neural Information Processing Systems, 34, 4207-4217.
> > > > > > > >
> > > > > > > >
> > > > > > > > [3] Locatello, F., Bauer, S., Lucic, M., Raetsch, G., Gelly, S., Schölkopf, B., & Bachem, O. (2019, May). Challenging common assumptions in the unsupervised learning of disentangled representations. In international conference on machine learning (pp. 4114-4124). PMLR.
> > > > > > > >
> > > > > > > > [4] Hyvarinen, A., Sasaki, H., & Turner, R. (2019, April). Nonlinear ICA using auxiliary variables and generalized contrastive learning. In The 22nd International Conference on Artificial Intelligence and Statistics (pp. 859-868). PMLR.
> > > > > > > >
> > > > > > > > [5] Pearl, J. (2009). Causal inference in statistics: An overview.
> > > > > > > >
> > > > > > > >
> > > > > > > > ---
> > > > > > > > Finally, thanks again for your insightful review and active discussion with us, we hope that this reply can help you better recognize our work and contribute to your discussion with other reviewers.

---

> > > > > > > > > ### Author Response · Authors · 2023-08-21
> > > > > > > > >
> > > > > > > > > Dear Reviewer Txp8:
> > > > > > > > >
> > > > > > > > > Since the rebuttal deadline is approaching rapidly, we would like to kindly inquire if we have adequately addressed your concerns. If there are more remaining issues, we would appreciate the chance to address them and work towards achieving a positive score.
> > > > > > > > >
> > > > > > > > > We deeply appreciate all the insightful comments you have posted, as they have greatly enhanced our paper. Simultaneously, we consider ourselves fortunate to have encountered such conscientious reviewers for our submission.
> > > > > > > > >
> > > > > > > > > Thanks

---

### Official Review · Reviewer_XaVg · 2023-07-08

**Soundness:** 3 good
**Presentation:** 3 good
**Contribution:** 3 good
**Rating:** 7
**Confidence:** 4

**Summary:**

The paper propose a novel learning framework OILCA for offline IL, which generate counterfactual data to augment the scarce expert data. They analyze the disentanglement identifiability of the constructed exogenous variable and the counterfactual identifiability of the augmented counterfactual expert data. The experiments especially in the CausalWorld benchmark demonstrate the effectiveness of the method.

**Strengths:**

1. It is in time to introduce causal inference to offline reinforcement learning and offline imitation learning. This topic is valuable to the RL community;
2. The paper is overall well-written and easy to follow for me;
3. The way of data augmentation through counterfactual inference makes sense;
4. The experiment is almost sufficient to demonstrate their method.


**Weaknesses:**

This paper can be improved in several areas:

Writing and structure:

1. In this paper, it's not clear what the spurious relations under the MDP structure are. Could you explain it based on Figure 2? Notably, it appears that there is a direct causal relationship between $u$ and $s$, which seems should not be categorized as spurious relations.

2. The definition of do-intervention from line 96 to 119 is a little confusing. Specifically, the definition of do on lines 96~96 and in Figure 2(b) is replacing $f$ with another function $\tilde f$ while keeping $u$, but on line 117, the interpretation of do changes to keeping the function $f$ unchanged and replacing $u$ with $\tilde u$. It seems that line 117 is more in line with the actual implementation. Could the author clarify this point?

3. The explanation in Section 2.3 on why IVAE solves the identifiability problem is a bit of vague. This is an important context, especially regarding the introduction of the auxiliary variable (which is c defined in Section 3.1) and its role in identifiability. It might be beneficial to move some content about c from section 3.1 to 2.3 to better inform the reader about the preliminaries.

Related work:

The authors do not mention any related work in the main body of the paper. Consider moving some of the related work from the appendix into the main text. Moreover, there have been several recent studies applying causal inference to data augmentation for RL . The authors can compare their method with these works and discuss the similarities or differences. They might also consider using these methods as baselines for comparison:
- Sample-Efficient Reinforcement Learning via Counterfactual-Based Data Augmentation
- MOCODA: Model-based Counterfactual Data Augmentation
- Counterfactual data augmentation using locally factored dynamics
- Offline Reinforcement Learning with Causal Structured World Models

Method:

1. The authors rely on a pretrained expert policy in the data augmentation process. How was this expert policy obtained? If we assume that we use $\mathcal{D}_E$ to train the expert policy, given that this paper claims $\mathcal{D}_E$ is very limited, it should be challenging to reconstruct the expert policy adequately from $\mathcal{D}_E$. Consequently, the constructed $\tilde s$, $\tilde a$ might be different from the actual data distribution in $\mathcal{D}_E$. How can we ensure that using the augmented dataset from $\tilde pi_e$, which is unreliable, will have a positive effect on the downstream imitation task?

2. The authors state that they ultimately use Equation (2) for data augmentation, but Equation (2) uses $p(u|c)$ to generate $u$, not the posterior-based method described on line 101, which should use $q(u|s,a,s',c)$ instead. The authors need to clarify the inconsistency between these two points.

3. The authors claim that their contribution is to prove that their method can improve the generalizability of imitation learning, which is a bit of overclaim. From their theoretical analysis, it only shows that increasing the amount of data can improve the accuracy of imitation learning, a relatively trivial conclusion that holds for any machine learning task. If we are conducting a theoretical analysis, we need to explain why using data generated by the counterfactual data augmentation method can better improve the algorithm's generalizability. Otherwise, it raises the question: Does data augmentation not reliant on counterfactual inference have the same improvement?

 Experiments:

1. To my knowledge, the original DEEPMIND CONTROL SUITE environment does not have exogenous variables. How did the authors construct the experimental environment and dataset to fit the setting they proposed? Supplementary note: if this is not achievable, I don't think experiments on MuJoCo are necessary. The authors could consider designing more experiments to validate their algorithm's effectiveness on benchmarks designed for causal inference, such as:
    - Alchemy: A benchmark and analysis toolkit for meta-reinforcement learning agents
    - Systematic Evaluation of Causal Discovery in Visual Model Based Reinforcement Learning

2. While the experimental results seem significant, the ablation study and visualizations are insufficient. Perhaps the authors could consider the differences between their data and data generated by different methods, the clustering of reconstructed exogenous variables, and whether the corresponding transitions or policy behaviors meet expectations.

3. The description of the baselines is insufficient. The authors should emphasize how the baselines utilize $\mathcal{D}_E$ and $\mathcal{D}_U$ respectively.

**Questions:**

see weaknesses

**Limitations:**

NAN

---

> ### Author Rebuttal · Authors · 2023-08-10
>
> Dear reviewer XaVg:
>
> Thanks for your review of our paper. Here's our response to the weaknesses, questions, and limitations you have highlighted:
>
> -------
>
> **Writing and structure:**
>
> **1.In this paper, it's not clear what the spurious relations under the MDP structure are.**
>
>
> **A**:  We follow an standard causal MDP setting [1][2][3], where both the relationship of state and the exogenous variable are the same as ours.
>
>
> [1] Kazemi, M., & Paoletti, N. (2022). Towards Causal Temporal Reasoning for Markov Decision Processes. arXiv preprint arXiv:2212.08712.
>
> [2] Oberst, M., & Sontag, D. (2019, May). Counterfactual off-policy evaluation with gumbel-max structural causal models. In International Conference on Machine Learning (pp. 4881-4890). PMLR.
>
> [3] Tsirtsis, S., De, A., & Rodriguez, M. (2021). Counterfactual explanations in sequential decision making under uncertainty. Advances in Neural Information Processing Systems, 34, 30127-30139.
>
> **2.The definition of do-intervention from line 96 to 119 is a little confusing.**
>
> **A**:  For this definition and the related figure are confused, we revise the Figure 2(b),
> https://i.postimg.cc/KzWk86kG/Figure2-b.jpg
>  and we will revise the related description in the final version.
>
> **3.The explanation in Section 2.3 on why IVAE solves the identifiability problem is a bit of vague.**
>
> **A**:  Thanks for the question, we will move part of the Section 3.1 into the preliminary.
>
> -------
>
> **Related Work:**
>
> **1.The authors do not mention any related work in the main body of the paper.**
>
>
> **A**:  The related work will be presented in the appendix due to space limitations. In the appendix, we will discuss the related work mentioned above [1][2][3]. Additionally, we have carefully examined [4], which proposes a method called FOCUS comprising two components: causal structure learning for a world model and offline model-based policy learning. In summary, the aforementioned related works are not suitable as baselines. [1] focuses on learning a personalized policy in a sequential decision process. [2,3] generate counterfactual samples using local causal models with factored MDP. [4] leverages causal discovery to improve the world model. However, in the final version, we will remove some related works from the main text.
>
> -------
> **Method:**
>
> **1.The authors rely on a pretrained expert policy in the data augmentation process. How was this expert policy obtained?**
>
> **A:** See the pre-trained policy answer (Q2) for Reviewer srX3.
>
> **2.Clarify the inconsistency between $p(u|c)$ and $p(u|s,a,c',c)$.**
>
> **A:** There is a mistake in the pseudocode when generating counterfactual states. Currently, the counterfactual states are generated using the posterior instead of the prior. We will correct this in the final version.
>
> **3.The authors claim that their contribution is to prove that their method can improve the generalizability of imitation learning, which is a bit of overclaim**
>
> **A:** For this question, we have presented the distribution of the counterfactual samples mentioned earlier. Drawing from the existing literature on counterfactual data augmentation [1][2], we aim to show that the samples generated by OILCA can be considered as $D_E$ with theoretical guarantees. Therefore, we analyze the outcomes from the standpoint of sample numbers.
> [1] MOCODA: Model-based Counterfactual Data Augmentation
> [2] Counterfactual data augmentation using locally factored dynamics
>
>
> -------
> **Experiment:**
>
> **1.The original DEEPMIND CONTROL SUITE environment does not have exogenous variables.**
>
> **A:** In the DEEPMIND CONTROL SUITE, different noise exists in the dynamic environment depending on the environment initialization. These variations can be considered as different distributions of the exogenous variable. Additionally, we utilize the DEEPMIND CONTROL SUITE because it is commonly employed in papers proposing these baselines. By comparing the results, we can assess the effectiveness of our training framework.
>
> As for the causal benchmark, we evaluate the performance using the causal world dataset. This benchmark is widely used for robotics control and closely aligns with the standard offline imitation learning setting.
>
> Furthermore, we explore the above-mentioned benchmarks for causal meta-reinforcement learning and model-based reinforcement learning. However, it may require some additional time to integrate them into the standard offline imitation learning setting, and the suitability of these datasets for offline IL needs to be further discussed.
>
> **2.While the experimental results seem significant, the ablation study and visualizations are insufficient.**
>
> **A:** To address the challenge of not having access to the true distribution of the exogenous variable, we visualize the learned exogenous variable alongside the true exogenous variable using a synthetic dataset. Furthermore, rather than conducting an ablation study, we prioritize compatibility as a training framework. This is because we believe that compatibility plays a more crucial role. Considering the limited research on data augmentation in offline imitation learning, combining different data augmentation methods is seen as a valuable contribution in this field. We intend to further discuss this aspect in our future work.
>
>
>
> **3.The description of the baselines is insufficient. The authors should emphasize how the baselines utilize $\mathcal{D}_E$ and $\mathcal{D}_U$ respectively.**
>
> **A:** We described the baselines in the appendix, to answer this question, we will add more detailed about how these methods use the expert and unlabeled data.

---

> > ### Author Response · Authors · 2023-08-19
> > **Additional response for weaknesses (Part1)**
> >
> > **Method W3**
> >
> > Thanks for your insightful question, we provide a new theoretical gaurantee for you to understand the problem, moreover, we analyse it from a supervised learning perspective:
> >
> > **Theoretical gaurantee**
> >
> > Let $X \subseteq \mathbb{R}^d$ be the input space and $y \subseteq \mathbb{R}$ be the label space. We denote by $\mathcal{D}$ the population distribution over $z=X \times \mathcal{Y}$. The $L\_p$ norm of a random variable $X$ is denoted as $\|X\|\_p=\left(\mathbb{E}|X|^p\right)^{\frac{1}{p}}$. Given a set $S= \{ \mathbf{z}\_1, \mathbf{z}\_2, \ldots, \mathbf{z}\_m\}$, we define $S^{\backslash i}$ as the set after removing the $i$-th data point in the set $S$, and $S^i$ as the set after replacing the $i$-th data point with $\mathbf{z}\_i^{\prime}$ in the set $S$. Let $[m]=\{1,2, \ldots, m\}$, then for every set $V \subseteq[n]$, we define $S\_V=\{\mathbf{z}\_i: i \in V\}$. In addition, for some function $f=f(S)$, we denote its conditional $L\_p$ norm with respect to $S\_V$ by $\|f\|\_p\left(S\_V\right)=\left(\mathbb{E}\left[\|f\|^p \mid S\_V\right]\right)^{\frac{1}{p}}$. Besides, we denote the total variation distance by $d\_{\mathrm{TV}}$ and KL divergence by $d\_{\mathrm{KL}}$, respectively.
> >
> > We let $(y)^x$ be the set of all measurable functions from $X$ to $y, \mathcal{A}$ be a learning algorithm and $\mathcal{A}(S) \in({y})^x$ be the hypothesis learned on the dataset $S$. Given a learned hypothesis $\mathcal{A}(S)$ and a loss function $\ell:(y)^x \times \mathcal{Z} \rightarrow \mathbb{R}\_{+}$, the true error $\mathcal{R}\_{\mathcal{D}}(\mathcal{A}(S))$ with respect to the data distribution $\mathcal{D}$ is defined as $\mathbb{E}\_{\mathbf{z} \sim \mathcal{D}}[\ell(\mathcal{A}(S), \mathbf{z})]$. In addition, the corresponding empirical error $\widehat{\mathcal{R}}\_S(\mathcal{A}(S))$ is defined as $\frac{1}{m} \sum\_{i=1}^m \ell\left(\mathcal{A}(S), \mathbf{z}\_i\right)$.
> >
> >
> > In this part, we describe the process of DA (Data augmentation) in a mathematical way. Given a training set $S$ with $m\_S$ i.i.d. examples from $\mathcal{D}$, we can train a data augmentation model $G$, and denote the model distribution by $\mathcal{D}\_G(S)$. We note that the randomness from training the augmentation model is ignored. In addition, we define the expectation of the model distribution with regard to $S$ as $\mathcal{D}\_G=\mathbb{E}\_S\left[\mathcal{D}\_G(S)\right]$. Based on the trained data augmentation, we can then obtain a new dataset $S\_G$ with $m\_G$ i.i.d. samples from $\mathcal{D}\_G(S)$, where $m\_G$ is a hyperparameter. Typically, we consider the case that $m\_G=\Omega\left(m\_S\right)$ if DA is utilized. We denote the total number of the data points in augmented set $\widetilde{S}=S \cup S\_G$ by $m\_T$. Besides, we define the mixed distribution after augmentation as $\widetilde{\mathcal{D}}(S)=\frac{m\_S}{m\_T} \mathcal{D}+\frac{m\_G}{m\_T} \mathcal{D}\_G(S)$. As a result, a hypothesis $\mathcal{A}(\widetilde{S})$ can be learned on the augmented dataset $\widetilde{S}$.
> >
> > To understand the effect of DA, we are focus on the generalization error $\left|\mathcal{R}\_{\mathcal{D}}(\mathcal{A}(\widetilde{S}))-\widehat{\mathcal{R}}\_{\widetilde{S}}(\mathcal{A}(\widetilde{S}))\right|$ with regard to the learned hypothesis $\mathcal{A}(\widetilde{S})$. For convenience, we denote it by Gen-error in the remaining part.
> >
> > **Definition 1 (Uniform stability)**. Algorithm $\mathcal{A}$ is uniformly $\beta\_m$-stable with respect to the loss function $\ell$ if the following holds
> > $$
> > \forall S \in Z^m, \forall \mathbf{z} \in Z, \forall i \in[m], \sup \_{\mathbf{z}}\left|\ell(\mathcal{A}(S), \mathbf{z})-\ell\left(\mathcal{A}\left(S^i\right), \mathbf{z}\right)\right| \leq \beta\_m
> > $$
> >
> > To understand DA, the generalization error of the hypothesis $\mathcal{A}(\widetilde{S})$ learned on the dataset $\widetilde{S}$ after augmentation. Formally, we need to bound $\left|\mathcal{R}\_{\mathcal{D}}(\mathcal{A}(\widetilde{S}))-\widehat{\mathcal{R}}\_{\widetilde{S}}(\mathcal{A}(\widetilde{S}))\right|$, which has been defined as Gen-error. Recall that $\widetilde{\mathcal{D}}(S)$ has been defined as the mixed distribution after augmentation, to derive such a bound, we first decomposed Gen-error as
> > $$
> > \mid \text { Gen-error } \mid \leq \underbrace{\left|\mathcal{R}\_{\mathcal{D}}(\mathcal{A}(\widetilde{S}))-\mathcal{R}\_{\widetilde{\mathcal{D}}(S)}(\mathcal{A}(\widetilde{S}))\right|}\_{\text {Distributions' divergence }}+\underbrace{\left|\mathcal{R}\_{\widetilde{\mathcal{D}}(S)}(\mathcal{A}(\widetilde{S}))-\widehat{\mathcal{R}}\_{\widetilde{S}}(\mathcal{A}(\widetilde{S}))\right|}\_{\text {Generaliztion error w.r.t. mixed distribution }} .
> > $$

---

> > > ### Author Response · Authors · 2023-08-19
> > > **Additional response for weaknesses (Part2)**
> > >
> > > The first term on the right hand can be bounded by the divergence (e.g., $d\_{\mathrm{TV}}, d\_{\mathrm{KL}}$ ) between the mixed distribution $\widetilde{\mathcal{D}}(S)$ and the true distribution $\mathcal{D}$. It is heavily dependent on the ability of the chosen augmentation model. For the second term, we note that classical stability bounds can not be used directly, because points in $\widetilde{S}$ are drawn non-i.i.d.. We mainly use a core property of $\widetilde{S}$, that is, $S$ satisfies the i.i.d. assumption, and $S\_G$ satisfies the conditional i.i.d. assumption when $S$ is fixed. Inspired by this property, we furthermore decompose this term and utilize sharp moment inequalities to obtain an upper bound. Finally, we conclude with the following result.
> > >
> > > **Theorem 1 (Corollary 8, [1])**. Assume that $\mathcal{A}$ is a $\beta\_m$-stable learning algorithm and the loss function $\ell$ is bounded by $M$. Given a training set $S$ with $m$ i.i.d. examples sampled from the distribution $\mathcal{D}$, then for any $\delta \in(0,1)$, with probability at least $1-\delta$, it holds that
> > > $$
> > > \left|\mathcal{R}\_{\mathcal{D}}(\mathcal{A}(S))-\widehat{\mathcal{R}}\_S(A(S))\right| \lesssim \log (m) \beta\_m \log \left(\frac{1}{\delta}\right)+M \sqrt{\frac{1}{m} \log \left(\frac{1}{\delta}\right)} .
> > > $$
> > > We note that all generalization bounds mentioned above require a primary condition: data points are drawn i.i.d. according to the population distribution $\mathcal{D}$. However, it no longer holds in the setting of DA. On the one hand, the distribution $\mathcal{D}\_G(S)$ learned by the augmentation model is generally not the same as the true distribution $\mathcal{D}$. On the other hand, the learned $\mathcal{D}\_G(S)$ is heavily dependent on the sampled dataset $S$. This property brings obstacles to the derivation of the generalization bound for DA.
> > >
> > > **Theorem 2 (Generalization bound for DA)**. Assume that $\mathcal{A}$ is a $\beta\_m$-stable learning algorithm and the loss function $\ell$ is bounded by $M$. Given an set $\widetilde{S}$ augmented, then for any $\delta \in(0,1)$, with probability at least $1-\delta$, it holds that
> > > $$
> > > \begin{aligned}
> > > \mid \text { Gen-error } \mid & \lesssim \underbrace{\frac{m\_G}{m\_T} M d\_{\mathrm{TV}}\left(\mathcal{D}, \mathcal{D}\_G(S)\right)}\_{\text {Distributions' divergence }}+\frac{M\left(\sqrt{m\_S}+\sqrt{m\_G}\right)+m\_S \sqrt{m\_G} \beta\_{m\_T}}{m\_T} \sqrt{\log \left(\frac{1}{\delta}\right)} \\\\
> > > & +\frac{\beta\_{m\_T}\left(m\_S \log m\_S+m\_G \log m\_G\right)+m\_S \log m\_S M \mathcal{T}\left(m\_S, m\_G\right)}{m\_T} \log \left(\frac{1}{\delta}\right),
> > > \end{aligned}
> > > $$
> > > where $\mathcal{T}\left(m\_S, m\_G\right)=\sup \_i d\_{\mathrm{TV}}\left(\mathcal{D}\_G^{m\_G}(S), \mathcal{D}\_G^{m\_G}\left(S^i\right)\right)$.
> > >
> > > [1] Olivier Bousquet, Yegor Klochkov, and Nikita Zhivotovskiy. Sharper bounds for uniformly
> > > stable algorithms. In COLT, volume 125, pages 610–626, 2020.
> > >
> > >
> > > From above analysis, we can see that our method will have a tighter generalization bound compared to the data augmentation policy.
> > >
> > > Moreover, it is common to use a weak expression model to enhance data and use a strong expression model to learn new algorithms in the field of data augmentation.  And we will add this in the final revised version.
> > >
> > > ----
> > >
> > > **Experiment**
> > >
> > > **W1**
> > >
> > > In the DeepMind Control Suite, during the collection of offline data, we apply random Gaussian perturbation to the action outputted by the policy. This perturbation is specified in the XML configuration file as an integral part of the environment. Additionally, the distribution of the perturbation differs across different environment initializations (auxiliary variable $c$) due to their initialization seeds. Especially, different seeds correspond to different mean and variance of the Gaussian distribution perturbation via the random number generator. This approach is employed to introduce uncertainty into the environment, thereby aligning with our problem setting.
> > >
> > > Finally, thanks for you careful reviews of our work, we will add the above contents to our revised version.

---

> > > > ### Comment · Reviewer_XaVg · 2023-08-19
> > > >
> > > > Thanks for your detailed response. Most of my concerns have been solved according to the rebuttal letter.

---

### Official Review · Reviewer_agsz · 2023-07-16

**Soundness:** 3 good
**Presentation:** 2 fair
**Contribution:** 3 good
**Rating:** 6
**Confidence:** 4

**Summary:**

This paper introduces OILCA, a causality-regularized data augmentation method for offline imitation learning tasks. Overall, the idea is novel to me. The empirical results shown in the experiment section seem very promising. However, to support the claims made in the paper, more experiments are needed. Please see details in my suggestions & questions section.

**Strengths:**

Overall, the idea is novel to me. The authors provide theoretical analysis to support their results. The empirical results shown in the experiment section seem very promising.

**Weaknesses:**

Missing reference:

As your work is related to causality + imitation learning, I believe the work of Causal confusion in imitation learning clearly worths a citation and discussion. Also, there is quite a lot of related work that is not discussed, e.g., with a quick Google search on causality + reinforcement learning:

[1] Pim de Haan, Dinesh Jayaraman, and Sergey Levine. Causal confusion in imitation learning. In NeurIPS, pages 11698–11709, 2019.

[2] Gasse, Maxime, et al. "Causal reinforcement learning using observational and interventional data." arXiv preprint arXiv:2106.14421, 2021.

[3] Sun, H., & Wang, T.. Toward Causal-Aware RL: State-Wise Action-Refined Temporal Difference. arXiv preprint arXiv:2201.00354, 2022.




Suggestions:

The flow of the introduction is a bit awkward to me: the authors mentioned the difficulty of learning from a mixed dataset containing both expert and random-quality data, but they then answer the counterfactual question of how the expert could perform under different states, this question is uncorrelated with the previous discussion. I hope the authors can update this part to better convey their idea and motivations.

The notions in Definition 2 are unclear to me. what does the \tilde mean when it is over PA_i? This is not explained in the paper.

Figure 5 is not mentioned in the text and is therefore confusing: how do you increase the percentage of expert data? Why is the proportion always smaller than 1.0?

It's sort of misleading to call a well-performing method a 'robust' method. (Q2, experiment section). To demonstrate the robustness, the authors should show far more empirical studies rather than a comparison under a single setting.

**Questions:**

1.**(Very Important) A critical confusion is that, if you can do accurate counterfactual reasoning over other states (not included in the expert trajectory), why do not use such a model directly perform as the learned policy?**
Actually, if I am right, this is a **very strong assumption** for your theoretical proof --- if you assume the counterfactual reasoning is accurate (such that the augmentation is reliable), you have already guaranteed to achieve an improved result.

2. How is your algorithm's sensitivity to hyper-parameters? This is especially important for offline RL/IL algorithms --- the application scenarios do not permit excessive hyper-parameter tuning, hence a one-for-all hyper-parameter is of great importance. I would like to see an ablation study on stress-testing not only OILCA but also other baseline methods.
The underlying question is: while causal augmentation is beneficial, does it add extra instability to the learning process?

3. On the performance gain: Can the authors disclose what settings have been changed over previous algorithms like ORIL and DWBC?
If the training settings are the same, the performance gain can be attributed to the augmented expert data, then what do other algorithms perform under this augmented dataset?

4. How much expert data is needed in augmentation? As it has been shown that Behavior Cloning under clean and high-quality data can perform as good as imitation learning[4], it will be great to show some results to demonstrate in a minimal effort that the data augmentation is effective. --- I'm keen to see under what specific setting, BC can be as powerful as those offline IL methods, and how your method's performance surpasses others when you gradually decrease the proportion of expert data.

[4] Mandlekar, Ajay, et al. "What matters in learning from offline human demonstrations for robot manipulation." arXiv preprint arXiv:2108.03298 (2021).

**Limitations:**

please see above.

---

> ### Author Rebuttal · Authors · 2023-08-10
>
> Dear reviewer asgz:
>
> Thanks for your review of our paper. Here's our response to the weaknesses, questions, and limitations you have highlighted:
>
> -------
> **Weaknesses:**
>
> **W1. Missing references.**
>
>
> **A**: Thank you for the valuable suggestion. We will include and discuss the raised related works in our revision of the paper.
>
> **W2. The flow of the introduction.**
>
> **A**: Thank you for the suggestion. We will update the introduction part to further illustrate our idea and motivation.
>
> For the limited expert data, we choose data augmentation to solve this problem, and with the advantage of counterfactual reasoning, we adopt the counterfactual data augmentation, which is to answer the counterfactual question.
>
>
>
> **W3. What does the \tilde mean when it is over PA_i?**
>
> **A**: $\mathbf{PA}_i$ is the parent of $X_i$. $\tilde{\mathbf{PA}}_i$ is the intervented parent of $X_i$. We will clarify it in our revision.
>
>
>
>
>
> **W4. Figure 5: How do you increase the percentage of expert data? Why is the proportion always smaller than 1.0?**
>
> **A**: Figure 5 presents the experiment analyzing the extent to which the number of augmented expert data influences the performance of the learned policy. This experiment is described in detail in the paper (lines 297-302).
>
> We increase the number of augmented expert samples and concanate them into $D_E$, until the proportion $|D_E|/|D_U|=1$. So the proportion is always smaller than or equal to 1.0.
>
>
>
>
> **W5. In distribution robustness?**
>
> **A**: This robustness is to demonstrate the performance of in distribution online testing, rather than the robustness for the additive noise.
> We apologize for any potential confusion and we will address and revise this in the final version.
>
>
> -------
>
> **Questions:**
>
>
>
> **Q1. (Very Important) Assumption: the counterfactual reasoning is accurate.**
>
> **A**: We thank the reiviewer for raising this concern.
>
> Even when the counterfactual reasoning process is not perfect,  our framework can provide a trade-off between the information brought by the augmented data and the causal model accuracy.
>
> Even when the counterfactual reasoning process is not perfect,  our framework can provide a trade-off between the information brought by the augmented data and the causal model accuracy.
>
> Experiments show that our framework can indeed bring additional performance gains.
>
>
> **Q2. Algorithm's sensitivity to hyper-parameters.**
>
>
> **A**:  Due to the space limitation here, we put our detailed rebuttal for this question into an external anonymous link (https://hackmd.io/@littleshoes/ryaYsJfhn). Please check the Q2 of Reviewer asgz in it.
>
>
> **Q3. What settings have been changed over previous algorithms like ORIL and DWBC?**
>
> **A**:  Due to the space limitation here, we put our detailed rebuttal for this question into an external anonymous link (https://hackmd.io/@littleshoes/ryaYsJfhn). Please check the Q3 of Reviewer asgz in it.
>
>
> **Q4.How much expert data is needed in augmentation?**
>
> **A**:  Due to the space limitation here, we put our detailed rebuttal for this question into an external anonymous link (https://hackmd.io/@littleshoes/ryaYsJfhn). Please check the Q4 of Reviewer asgz in it.

---

> > ### Author Response · Authors · 2023-08-10
> > **Revisement to the Response Above**
> >
> > **1. A Revised version of our response to Q1**
> >
> >  **Q1. (Very Important) Assumption: the counterfactual reasoning is accurate.**
> >
> > **A**: We thank the reviewer for raising this concern.
> >
> > Even when the counterfactual reasoning process is not perfect,  our framework can provide a trade-off between the information brought by the augmented data and the causal model accuracy.
> >
> > If the benefits introduced by the augmented data are larger than the harm brought by the model errors, then our framework can overall bring effectiveness.
> >
> > Experiments show that our framework can indeed bring additional performance gains.
> >
> > **2. A little typo: the Reviewer asgz above should be agsz.**
> >
> > Thanks again for your careful reviews. Looking forward to your feedback and further discussion.

---

> > > ### Comment · Reviewer_agsz · 2023-08-11
> > > **Thank you for response**
> > >
> > > Many thanks to the authors for their explanations.
> > >
> > > 1. I think my question 1 is not directly answered: if you can do accurate counterfactual reasoning over other states (not included in the expert trajectory), why do not use such a model directly perform as the learned policy?
> > >
> > > 2. "Even when the counterfactual reasoning process is not perfect, our framework can provide a trade-off between the information brought by the augmented data and the causal model accuracy."
> > >
> > > An incremental study can be helpful here to show how much performance is gained using the generated data, compared to the baseline I mentioned: directly using your counterfactual reasoning over other states as the "policy".
> > >
> > > 3. "line 297-300"
> > > Those descriptions are not clear. "We increase the number of augmented expert samples and ...", still, why must this be smaller than 1?
> > >
> > > In practice, how do you determine the number of augmented data? e.g., in different tasks, how to guarantee the performance is not decaying?
> > >
> > >
> > > ---
> > > I understand you may need extra characters to explain your point. Yet the author's response **should not** contain any links, according to the code of conduct.

---

> > > > ### Author Response · Authors · 2023-08-14
> > > > **Response to Reviewer agsz's Follow-up Questions (Part 1)**
> > > >
> > > > Dear Reviewer agsz:
> > > >
> > > > Thanks for your follow-up questions. To address your concerns and help you better understand this work, we make the corresponding explanations and clarifications as follows.
> > > >
> > > > -------
> > > > **Q1**: (Very Important) Assumption: the counterfactual reasoning is accurate.
> > > >
> > > > **A**:
> > > > To better alleviate your concerns, we answer your question from both theoretical (**A1-1**) and empirical (**A1-2**) perspectives.
> > > >
> > > > Before diving into these details, we first review the process of counterfactual reasoning at a high level. That is, our counterfactual reasoning procedure first uses an identified conditional VAE to predict the $\tilde{s}\_{t+1}$. Then to generate the $\tilde{a}\_{t+1}$, we use a pretrained policy $\hat{\pi}\_E$ to interact with this $\tilde{s}\_{t+1}$.
> > > >
> > > > In **A1-1**, **from the theoretical perspective**, we introduce a new theory based on PAC learning, which reveals the relation between the error of the counterfactual reasoning process and the number of generated samples. In general, increasing the number of generated samples can help reduce the negative impact brought by the couterfactual reasoning errors. This demontrates that our method does not rely on the accurate counterfactual reasoning theoretically.
> > > >
> > > > In **A1-2**, **from the empirical perspective**, we follow your advice and directly perform the pretrained policy in the testing environment. The empirical results show that it is indeed not well-performing enough. The performance distance between it and our OILCA also demonstrates the effectiveness of counterfactual data augmentation module.
> > > >
> > > >
> > > > **A1-1.** The theorical analysis of the trade-off between the augmented samples and the pre-trained policy error.
> > > >
> > > > **Theorem 1.** Given a hypothesis class $\mathcal{H}$, for any $\epsilon, \delta \in(0,1)$ and $\eta \in(0,0.5)$, if $\hat{\pi}\_E \in \mathcal{H}$ is the pretrained policy model learned based on the empirical risk minimization (ERM), and the sample complexity (i.e., number of samples) is larger than $\frac{2 \log \left(\frac{2|\mathcal{H}|}{\delta}\right)}{\epsilon^2(1-2 \eta)^2}$, then the error between the model estimated and true results is smaller than $\epsilon$ with probability larger than $1-\delta$.
> > > >
> > > >
> > > > *Proof.* Suppose the prediction error of $\hat{\pi}\_E$ is $e$ (i.e., $\sum \mathbb{I}(a\_t^{\hat{\pi}\_E} \neq a\_t )=e), a\_t$ is the true action that an expert take, then the mis-matching probability between the observed and predicted results comes from two parts: (1) The observed result is true, but the prediction is wrong, that is, $e(1-\eta)$. (2) The observed result is wrong, but the prediction is right, that is $(1-e) \eta$. Thus, the total mis-matching probability is $\eta+e(1-2 \eta)$.
> > > >
> > > > The following proof is based on the reduction to absurdity. We firstly propose an assumption, and then derive contradicts to invalidate the assumption.
> > > >
> > > > - Assumption. Suppose the prediction error of $\hat{\pi}\_E$ (i.e., $e$ ) is larger than $\epsilon$, Then, at least one of the following statements hold: (1) The empirical mis-matching rate of $\hat{\pi}\_E$ is smaller than $\eta+\frac{\epsilon(1-2 \eta)}{2}$. (2) The empirical mis-matching rate of the optimal $h^* \in \mathcal{H}$ (i.e., the prediction error of $h^*$ is 0$)$ is larger than $\eta+\frac{\epsilon(1-2 \eta)}{2}$. These statements are easy to understand, since if both of them do not hold, we can conclude that the empirical loss of $\hat{\pi}\_E$ is larger than that of $h^*$, which does not agree with the ERM definition.
> > > > - Contradicts. To begin with, we review the uniform convergence properties [1] by the following lemma:
> > > >
> > > > **Lemma 1.** Let $\mathcal{H}$ be a hypothesis class, then for any $\epsilon \in(0,1)$ and $h \in \mathcal{H}$, if the number of training samples is $m$, the following formula holds:
> > > > $$
> > > > \mathbb{P}(|R(h)-\hat{R}(h)|>\epsilon)<2|\mathcal{H}| \exp \left(-2 m \epsilon^2\right)
> > > > $$
> > > > where $R$ and $\hat{R}$ are the expectation and empirical losses, respectively.

---

> > > > > ### Author Response · Authors · 2023-08-14
> > > > > **Response to Reviewer agsz's Follow-up Questions (Part 2)**
> > > > >
> > > > >
> > > > >
> > > > > For statement (1), since the prediction error of $\hat{\pi}\_E$ is larger than $\epsilon$, the expectation loss $R(\hat{\pi}\_E)$ is larger than $\eta+\epsilon(1-2 \eta)$. If the empirical loss $\hat{R}(\hat{\pi}\_E)$ is smaller than $\eta+\frac{\epsilon(1-2 \eta)}{2}$, then $|R(\hat{\pi}\_E)-\hat{R}(\hat{\pi}\_E)|$ should be larger than $\frac{\epsilon(1-2 \eta)}{2}$. At the same time, according to Lemma 1, when the sample number $m$ is larger than $\frac{2 \log \left(\frac{2|\mathcal{H}|}{\delta}\right)}{\epsilon^2(1-2 \eta)^2}$, we have $\mathbb{P}\left(|R(\hat{\pi}\_E)-\hat{R}(\hat{\pi}\_E)|>\frac{\epsilon(1-2 \eta)}{2}\right)<\delta$.
> > > > >
> > > > > For statement (2), the expectation loss of $h^*$ is $\eta$, i.e., $R\left(h^*\right)=\eta$. If the empirical loss $\hat{R}\left(h^*\right)$ is larger than $\eta+\frac{\epsilon(1-2 \eta)}{2}$, then $\mid R\left(h^*\right)-$ $\hat{R}\left(h^*\right) \mid$ should be larger than $\frac{\epsilon(1-2 \eta)}{2}$. According to Lemma 1, when the sample number $m$ is larger than $\frac{2 \log \left(\frac{2|\mathcal{H}|}{\delta}\right)}{\epsilon^2(1-2 \eta)^2}$, we have $\mathbb{P}\left(\left|R\left(h^*\right)-\hat{R}\left(h^*\right)\right|>\frac{\epsilon(1-2 \eta)}{2}\right)<\delta$.
> > > > >
> > > > > As a result, both of the above statements hold with the probability smaller than $\delta$, which implies that the prediction error of $\hat{\pi}\_E$ is smaller than $\epsilon$ with the probability larger than $1-\delta$.
> > > > >
> > > > >
> > > > > In summary, from this theory, we can see: in order to guarantee same performance with a given probability (i.e., $\epsilon$ and $\delta$ are fixed), one needs to generate more than $\frac{2 \log \left(\frac{2|\mathcal{H}|}{\delta}\right)}{\epsilon^2(1-2 \eta)^2}$ sequences. If the noise level $\eta$ is larger, more samples have to be generated. Extremely, if the pretrained policy can only produce fully noisy information (i.e., $\eta=0.5$), then infinity number of samples are required, which is impossible in realities.
> > > > >
> > > > > **This theory reveals the relation between the number of generated samples and the potential noisy information (due to the inaccurate counterfactual reasoning) contained in them, that is increasing the number of generated samples can help reduce the negative impact brought by the couterfactual reasoning errors. We hope this can help you understand our framework in theory.**
> > > > >
> > > > > [1] Shai Shalev-Shwartz and Shai Ben-David. 2014. Understanding machine learning: From theory to algorithms. Cambridge university press
> > > > >
> > > > >
> > > > >
> > > > > **A1-2.** Incremental study on the performance gain of our method over the pre-trained policy used to conduct the counterfactual reasoning.
> > > > >
> > > > > We follow your suggestions and directly take the pre-trained policy used to conduct the counterfactual reasoning as the baseline.
> > > > >
> > > > > The experiment results on Deepmind Control Suite (corresponding to Table 1 in the paper):
> > > > >
> > > > > |Task Name|Pre-trained Policy|OILCA|
> > > > > |------------------|------------------|------------------|
> > > > > |Cartpole Swingup|236.52 $\pm$ 17.19|608.38 $\pm$ 35.54|
> > > > > |Cheetah Run|66.59 $\pm$ 11.09|116.05 $\pm$ 14.65|
> > > > > |Finger Turn Hard|163.57 $\pm$ 11.46|298.73 $\pm$ 5.11|
> > > > > |Fish Swim|91.29 $\pm$ 8.14|290.28 $\pm$ 10.07|
> > > > > |Humanoid Run|90.73 $\pm$ 13.21|460.96 $\pm$ 17.55|
> > > > > |Manipulator Insert Ball|86.30 $\pm$ 9.25|296.83 $\pm$ 3.43|
> > > > > |Manipulator Insert Peg|124.38 $\pm$ 12.77|305.66 $\pm$ 4.91|
> > > > > |Walker Stand|178.16 $\pm$ 13.20|395.51 $\pm$ 8.05|
> > > > > |Walker Walk|58.25 $\pm$ 9.76 |377.19 $\pm$ 15.87|
> > > > >
> > > > >
> > > > > The experiment results on CausalWorld (corresponding to Table 2 in the paper):
> > > > >
> > > > >
> > > > >
> > > > >
> > > > >
> > > > > |Task Name| Pre-trained Policy|OILCA|
> > > > > |------------------|------------------|------------------|
> > > > > |Reaching|317.63 $\pm$ 13.21|976.60 $\pm$ 20.13|
> > > > > |Pushing|298.78 $\pm$ 21.60|405.08 $\pm$ 24.03|
> > > > > |Picking|331.42 $\pm$ 5.74|491.09 $\pm$ 6.44|
> > > > > |Pick and Place|317.45 $\pm$ 12.31|490.24 $\pm$ 11.69|
> > > > > |Stacking2|464.31 $\pm$ 15.82|831.82 $\pm$ 11.78|
> > > > > |Towers|704.59 $\pm$ 22.67|994.82 $\pm$ 5.76|
> > > > > |Stacked Blocks|659.28 $\pm$ 17.60|2617.71 $\pm$ 88.07|
> > > > > |Creative Stacked Blocks|584.25 $\pm$ 31.37|1348.49 $\pm$ 55.05|
> > > > > |General|529.14 $\pm$ 25.08|891.14 $\pm$ 23.12|
> > > > >
> > > > >
> > > > > From the tables, we can easily notice the performance of the pre-trained policy is still far inferior to our method OILCA. This demonstrates the effectiveness of our counterfactual data augmentation module.

---

> > > > > > ### Author Response · Authors · 2023-08-14
> > > > > > **Response to Reviewer agsz's Follow-up Questions (Part 3)**
> > > > > >
> > > > > > **Q4.** Clarification to the number of augmented data.
> > > > > >
> > > > > > **A:** We first apologize for any confusion brought by our line 297-300 in the paper. Here, we make the further clarification. Yes, the proportion of $D\_E$ and $D\_U$ is not necessarily smaller than 1. In fact, we can generate any amount of augmented expert data. So, this proportion can be any number, smaller than 1, equal to 1, or larger than 1. All of them are achievable. The point we want to hightlight with Figure 5 is that within a centain interval (usually smaller than 1), improving this proportion ($D\_E$/$D\_U$) can improve the performance, while the performance will converge when the proportion is great enough (usually larger than 1). This can be easily concluded from Figure 5 and the tables we provided in our previous rebuttal. **So, based on this empirical observation, we set this proportion as 1 in practice and all our experiments in this paper (like Table 1 and 2)**. Here, to prove that the performance will not decay when further improving the $D\_E$/$D\_U$, we further increase $D\_E$/$D\_U$ (larger than 1) and conduct the experiments to our method OILCA. The results are as follows.
> > > > > >
> > > > > > |Proportion|Cartpole Swingup|Cheetah Run|Finger Turn Hard|
> > > > > > |-------------------------------------|------------------|------------------|------------------|
> > > > > > |Original proportion (no augmentation， <10%)|382.55  $\pm$  8.95|66.87  $\pm$  4.60|243.47  $\pm$  17.12|
> > > > > > |10%|430.21  $\pm$  13.20|71.85  $\pm$  8.26|261.77  $\pm$  14.68|
> > > > > > |30%|463.78  $\pm$  21.95|86.44  $\pm$  13.62|269.85  $\pm$  13.39|
> > > > > > |50%|502.81  $\pm$  20.76|92.60  $\pm$  16.51|276.12  $\pm$  9.82|
> > > > > > |70%|557.90  $\pm$  16.62|105.57  $\pm$  11.29|283.69  $\pm$  12.71|
> > > > > > |90%|589.01  $\pm$  38.29|113.12  $\pm$  9.25|288.27  $\pm$  7.09|
> > > > > > |100%|608.38  $\pm$  35.54|116.05  $\pm$  14.65|298.73  $\pm$  5.11|
> > > > > > |200%|596.52 $\pm$ 28.37|106.39 $\pm$ 10.08|303.64 $\pm$ 12.91|
> > > > > > |300%|612.30 $\pm$ 41.25|118.51 $\pm$ 15.72|301.57 $\pm$ 8.30|
> > > > > > |500%|601.47 $\pm$ 27.82|109.96 $\pm$ 9.84|289.15 $\pm$ 15.27|
> > > > > > |1000%|605.81 $\pm$ 31.63|117.08 $\pm$ 7.69|295.48 $\pm$ 7.84|
> > > > > >
> > > > > >
> > > > > > Due to space limitation, we only provide the results on three tasks above. The trends on other tasks are similar. From the table, we can find that the performance will converge when the proportion is close to 1, and further improving it indeed will not influence the performance obviously. This can be explained that augmenting too much data can hardly bring additional effective information gain to the policy. And the performance of OILCA is inherently upper bounded by the optimal policy (Theorem 4).
> > > > > >
> > > > > > ------
> > > > > > Besides, thank you for your kind reminder and we apologize for the improper use of the external link. We indeed did not notice this point in the rebuttal policy. But we argue that our provided external is absolutely anonymous, so we didn't violate the anonymity principle. Here, we promise we will not use any external link in the following discussion. Thanks again for your careful reviews, insightful questions, and kind suggestions. Hope our above response can help address your concern. We are looking forward to the futher discussion if any follow-up questions.

---

> ### Comment · Reviewer_agsz · 2023-08-18
> **Follow-up question**
>
> I sincerely appreciate the authors' response and their diligent work in providing additional empirical evidence.
>
> According to the current results, the counterfactual reasoning policy does not perform well, indicating the augmented data does not perfectly
>
> To make the ablation study concrete, I believe another important experiment is to compare the performance of BC + augmented data. In this way, the performance gain of each element can be transparent. As has been shown in [1] (which is cited in your paper), BC is able to achieve on-par performance with IL with high-quality data.
>
> If time permitted, I would also see another important comparison: the performance gap when increasing expert data proportion using 1. the golden expert; and 2. using your method.
>
> I understand the discussion period is going to the end, if the authors could kindly provide an initial experiment on one or two environments, it will be much appreciated. And I believe those results will help a lot for further discussion among reviewers. Those experiments will provide general readers with a more comprehensive understanding of the performance (ability, pitfall) of your work and further enhance its impact.
>
> ---
> _**Reference**_
>
> [1] Mandlekar, Ajay, et al. "What matters in learning from offline human demonstrations for robot manipulation." arXiv preprint arXiv:2108.03298 (2021).

---

> > ### Author Response · Authors · 2023-08-19
> > **Additional Experiments (Part 1)**
> >
> > Dear Reviewer agsz:
> >
> >   Thanks for your appreciation and encouragement to our response. Here, to make the ablation study more concrete and better demonstrate the effectiveness of our proposed method, we follow your suggestions to conduct the following supplementary experiments.  Please forgive us for the limited additional experiments due to the limited time. We have tried our best to accomplish it.
> >
> >
> > ---------
> > **Additional Experiment 1:**
> > In fact, our experiment results for BC+augmented data have been provided in our response to Q3 in our initial rebuttal. In detail, it is Table 6 in that external link. We understand you may have neglected it due to the external link. Here, we provide it again as follows. (CA indicates the counterfactual data augmentation)
> > | Task Name| BC-exp| BC-exp+CA| BC-all| BC-all+CA| ORIL| ORIL+CA| BCND| BCND+CA| LobsDICE| LobsDICE+CA| DWBC| OILCA(DWBC+CA)|
> > |------------------|------------------|------------------|------------------|------------------|------------------|------------------|------------------|------------------|------------------|-------------------|------------------|------------------|
> > |Cartpole Swingup| 195.44  $\pm$  7.39  | 367.31  $\pm$  13.28 | 269.03  $\pm$  7.06  | 436.85  $\pm$  17.02 | 221.21  $\pm$  14.49 | 426.79  $\pm$  12.09 | 243.52  $\pm$  11.33 | 452.68  $\pm$  12.86 | 292.96  $\pm$  11.05 | 398.27  $\pm$   21.34 | 382.55  $\pm$  8.95  | 608.38  $\pm$  35.54 |
> > | Cheetah Run      | 66.59  $\pm$  11.09  | 94.73  $\pm$  8.22   | 90.0  $\pm$  31.74   | 125.19  $\pm$  18.20 | 45.08  $\pm$  9.88   | 78.44  $\pm$  6.95   | 96.06  $\pm$  16.15  | 158.62  $\pm$  8.85  | 74.53  $\pm$  7.75   | 89.65  $\pm$  12.04   | 66.87  $\pm$  4.60   | 116.05  $\pm$  14.65 |
> > | Finger Turn Hard | 129.20  $\pm$  4.51  | 186.97  $\pm$  17.46 | 104.56  $\pm$  8.32  | 152.38  $\pm$  11.67 | 185.57  $\pm$  26.75 | 227.94  $\pm$  15.47 | 204.67  $\pm$  13.18 | 284.29  $\pm$  12.03 | 190.93  $\pm$  12.19 | 237.83  $\pm$  24.91  | 243.47  $\pm$  17.12 | 298.73  $\pm$  5.11  |
> > | Fish Swim        | 74.59  $\pm$  11.73  | 164.35  $\pm$  12.91 | 68.87  $\pm$  11.93  | 137.98  $\pm$  6.74  | 84.90  $\pm$  1.96   | 156.92  $\pm$  8.18  | 153.28  $\pm$  19.29 | 268.56  $\pm$  6.03  | 188.84  $\pm$  11.28 | 229.24  $\pm$  13.62  | 212.39  $\pm$  7.62  | 290.29  $\pm$  10.07 |
> > | Reaching         | 281.18  $\pm$  16.45 | 608.77  $\pm$  15.42 | 176.54  $\pm$  9.75  | 527.61  $\pm$  10.58 | 339.40  $\pm$  12.98 | 652.21  $\pm$  7.05  | 228.33  $\pm$  7.14  | 582.44  $\pm$  9.07  | 243.29  $\pm$  9.84  | 461.38  $\pm$   14.05 | 479.92  $\pm$  18.75 | 976.60  $\pm$  20.13 |
> > | Pushing          | 256.64  $\pm$  12.70 | 343.80  $\pm$  9.79  | 235.58  $\pm$  10.23 | 356.17  $\pm$  13.81 | 283.91  $\pm$  19.72 | 367.46  $\pm$  6.31  | 191.23  $\pm$  12.64 | 320.94  $\pm$  10.37 | 206.44  $\pm$  15.35 | 263.74  $\pm$  12.84  | 298.09  $\pm$  14.94 | 405.08  $\pm$  24.03 |
> > | Picking          | 270.01  $\pm$  13.13 | 391.55  $\pm$  12.07 | 258.54  $\pm$  16.53 | 415.39  $\pm$  14.42 | 388.15  $\pm$  19.21 | 458.03  $\pm$  13.95 | 221.89  $\pm$  7.68  | 486.32  $\pm$  8.03  | 337.78  $\pm$  12.09 | 439.16  $\pm$  18.46  | 366.26  $\pm$  8.77  | 491.09  $\pm$  6.44  |
> > | Pick and Place   | 294.0  $\pm$  7.34   | 385.16  $\pm$  9.34  | 225.42  $\pm$  12.44 | 351.27  $\pm$  11.21 | 270.75  $\pm$  14.87 | 372.18  $\pm$  10.74 | 259.12  $\pm$  8.01  | 393.59  $\pm$  7.81  | 266.09  $\pm$  10.31 | 357.11  $\pm$  16.28  | 349.66  $\pm$  7.39  | 490.24  $\pm$  11.69 |
> >
> > From the table, we can observe that both the BC-exp+CA and BC-all+CA behave worse than our OILCA in most tasks. We attribute the reason for this phenomenon to that the performance of BC is more sensitive to the quality of training data, though it has the potential to achieve on-par performance with IL with  sufficient high-quality data. As discussed above, the counterfactual reasoning policy does not perform well, which means the augmented data is actually of uneven quality, still leaving a distance from the high-quality data. On the other hand, the BC-exp+CA and BC-exp+all both perform much better than vanilla BC-exp and BC-all, respectively. This demonstrates that our counterfactual data augmentation module can indeed consistently improve the imitation learning performance, regardless of the base method selection.
> >
> > --------

---

> > > ### Author Response · Authors · 2023-08-19
> > > **Additional Experiments (Part 2)**
> > >
> > > **Additional Experiment 2:**
> > > The performance gap when increasing expert data proportion using two kinds of augmented data: 1) sampling with golden expert policy in online environment, 2) our counterfactual data augmentation method. The results are as follows:
> > >
> > > **Cartpole Swingup Task**:
> > >
> > > |Proportion|Counterfactual Data Augmentation|Golden Expert|
> > > |-------------------------------------|------------------|------------------|
> > > |Original proportion (no augmentation， <10%)|382.55  $\pm$  8.95|382.55  $\pm$  8.95|
> > > |10%|430.21  $\pm$  13.20|441.36 $\pm$ 12.01|
> > > |30%|463.78  $\pm$  21.95|472.92 $\pm$ 7.62|
> > > |50%|502.81  $\pm$  20.76|520.15 $\pm$ 15.43|
> > > |70%|557.90  $\pm$  16.62|562.89 $\pm$ 20.47|
> > > |90%|589.01  $\pm$  38.29|593.37 $\pm$ 16.81|
> > > |100%|608.38  $\pm$  35.54|621.80 $\pm$ 9.26|
> > > |200%|596.52 $\pm$ 28.37|634.12 $\pm$ 18.29|
> > > |300%|612.30 $\pm$ 41.25|635.93 $\pm$ 25.15|
> > > |500%|601.47 $\pm$ 27.82|627.47 $\pm$ 22.86|
> > > |1000%|605.81 $\pm$ 31.63|629.94 $\pm$ 23.28|
> > >
> > > **Cheetah Run Task**:
> > >
> > > |Proportion|Counterfactual Data Augmentation|Golden Expert|
> > > |-------------------------------------|------------------|------------------|
> > > |Original proportion (no augmentation， <10%)|66.87  $\pm$  4.60|66.87  $\pm$  4.60|
> > > |10%|71.85  $\pm$  8.26| 74.56 $\pm$ 3.29|
> > > |30%|86.44  $\pm$  13.62| 82.06 $\pm$ 9.36 |
> > > |50%|92.60  $\pm$  16.51| 89.21 $\pm$ 12.98 |
> > > |70%|105.57  $\pm$  11.29| 111.27 $\pm$ 11.56 |
> > > |90%|113.12  $\pm$  9.25| 118.32 $\pm$ 15.27|
> > > |100%|116.05  $\pm$  14.65| 128.07 $\pm$ 8.31|
> > > |200%|106.39 $\pm$ 10.08|132.64 $\pm$ 14.24|
> > > |300%|118.51 $\pm$ 15.72|125.18 $\pm$ 8.73|
> > > |500%|109.96 $\pm$ 9.84| 129.72 $\pm$ 12.34|
> > > |1000%|117.08 $\pm$ 7.69| 124.80 $\pm$ 9.46|
> > >
> > > **Finger Turn Hard Task**:
> > >
> > > |Proportion|Counterfactual Data Augmentation|Golden Expert|
> > > |-------------------------------------|------------------|------------------|
> > > |Original proportion (no augmentation， <10%)|243.47  $\pm$  17.12|243.47  $\pm$  17.12|
> > > |10%|261.77  $\pm$  14.68|255.62 $\pm$ 18.29|
> > > |30%|269.85  $\pm$  13.39|272.18 $\pm$ 12.25|
> > > |50%|276.12  $\pm$  9.82|285.48 $\pm$ 8.36|
> > > |70%|283.69  $\pm$  12.71|295.83 $\pm$ 13.48|
> > > |90%|288.27  $\pm$  7.09|306.26 $\pm$ 10.81|
> > > |100%|298.73  $\pm$  5.11|303.51 $\pm$ 11.67|
> > > |200%|303.64 $\pm$ 12.91|311.70 $\pm$ 9.74|
> > > |300%|301.57 $\pm$ 8.30|305.42 $\pm$ 14.53|
> > > |500%|289.15 $\pm$ 15.27|302.15 $\pm$ 12.16|
> > > |1000%|295.48 $\pm$ 7.84|304.93 $\pm$ 11.19|
> > >
> > > From tables above, we can find that our counterfactual data augmentation still behaves slightly worse than augmentation with golden expert under most proportions, though achieving obvious improvement over other IL baselines. This is actually reasonable, because the augmented data sampled with the golden expert can be considered to be high-quality enough. Thus, the performance with such augmented data from golden expert can be regarded as a kind of upper bound for the data augmentation research in offline IL area.
> > >
> > > -----------
> > >
> > > Thanks again for your appreciation, insightful review and valuable suggestions to our work. Honestly hope the above empirical evidences can help you better recognize our work and contribute to your discussion with other reviewers.

---

> > > > ### Comment · Reviewer_agsz · 2023-08-21
> > > > **Raising Score**
> > > >
> > > > I would express my sincere appreciation for the authors' very detailed responses and diligent work during the discussion period!
> > > >
> > > > I increased my overall rating from 4 to 6, and my soundness score from 2 to 3, accordingly.
> > > >
> > > > I would suggest the authors provide full results during future revisions. I believe those results are helpful to general readers and beneficial to improving the soundness and impact of this work.

---

> > > > > ### Author Response · Authors · 2023-08-21
> > > > >
> > > > > Dear Reviewer agsz:
> > > > >
> > > > >    Thanks so much for your appreciation and encouragement.  Your insightful suggestions and careful reviews help improve the quality of our work a lot! We will definitely provide full results during future revisions according to your kind suggestion.  Besides, we will try our best to improve the soundness and impact of our work to help general readers better understand it.

---

### Decision · Program_Chairs · 2023-09-21

**Decision:**

Accept (poster)

**Comment:**

This paper presents OILCA, a causality-regularized data augmentation method for offline imitation learning. The proposed framework generates counterfactual data to augment expert data and demonstrates effectiveness in experiments, particularly in the CausalWorld benchmark. The idea looks interesting and novel, and the proposed method is sound with theoretical and experimental support.